# PREDICTION ERROR-BASED CLASSIFICATION FOR CLASS-INCREMENTAL LEARNING

**Michał Zając**
KU Leuven, ESAT-PSI
Jagiellonian University

**Tinne Tuytelaars**
KU Leuven, ESAT-PSI

**Gido M. van de Ven**
KU Leuven, ESAT-PSI

## ABSTRACT

Class-incremental learning (CIL) is a particularly challenging variant of continual learning, where the goal is to learn to discriminate between all classes presented in an incremental fashion. Existing approaches often suffer from excessive forgetting and imbalance of the scores assigned to classes that have not been seen together during training. In this study, we introduce a novel approach, Prediction Error-based Classification (PEC), which differs from traditional discriminative and generative classification paradigms. PEC computes a class score by measuring the prediction error of a model trained to replicate the outputs of a frozen random neural network on data from that class. The method can be interpreted as approximating a classification rule based on Gaussian Process posterior variance. PEC offers several practical advantages, including sample efficiency, ease of tuning, and effectiveness even when data are presented one class at a time. Our empirical results show that PEC performs strongly in single-pass-through-data CIL, outperforming other rehearsal-free baselines in all cases and rehearsal-based methods with moderate replay buffer size in most cases across multiple benchmarks.[1]

## 1 INTRODUCTION

Continual learning addresses the problem of incrementally training machine learning models when the data distribution is changing (Parisi et al., 2019; De Lange et al., 2022; van de Ven et al., 2022). Humans excel in learning incrementally from real-world data that are often noisy and non-stationary; embedding a similar continual learning capability in machine learning algorithms could make them more robust and efficient (Hadsell et al., 2020; Kudithipudi et al., 2022). A particularly challenging variant of continual learning is class-incremental learning (CIL) (Rebuffi et al., 2017; Belouadah et al., 2021; van de Ven et al., 2022; Masana et al., 2023), which involves jointly discriminating between all classes encountered during a sequence of classification tasks.

The most common approach for classification, particularly in deep learning, is to learn a discriminative classifier, as shown in Figure 1a. However, in continual learning scenarios, issues such as forgetting relevant features of past classes arise. To address this, discriminative classification can be supplemented with continual learning techniques such as replay (Chaudhry et al., 2019b; Lopez-Paz & Ranzato, 2017) or regularization (Kirkpatrick et al., 2017; Zenke et al., 2017). Nonetheless, these methods typically perform worse with class-incremental learning than with task-incremental learning (van de Ven et al., 2022), and they often suffer from imbalances between classes that have not been presented together during training (Wu et al., 2019). Recently, an alternative paradigm of generative classification (see Figure 1b) has been explored for CIL, with promising results (van de Ven et al., 2021; Banayeeanzade et al., 2021). This approach involves training a likelihood-based generative model for each class, and using class-conditional likelihoods as scores for classification during inference. However, learning decent generative models is difficult, especially for complex distributions and with limited training data. In this paper, we put forward a hypothesis that the generative modeling task is not necessary and that it is possible to obtain accurate scores from class-specific modules in more efficient ways by forgoing the generative capability.

As a realization of this idea, we propose Prediction Error-based Classification (PEC), depicted in Figure 1c. In this new approach to classification, we replace the generative modeling objective with a much simpler one, defined by a frozen random neural network. More precisely, for each class $c$, we train a neural network $g_{\theta^c}$ *exclusively on examples from that class*, to replicate the outputs of a frozen target random neural network $h$. During testing, the prediction error of $g_{\theta^c}$ on an example $x$ is utilized to generate a *class score* for class $c$, analogous to logits or likelihoods, used in discriminative and generative classification, respectively.

---

[1]We release source code for our experiments: `https://github.com/michalzajac-ml/pec`

This approach is inspired by Random Network Distillation (Burda et al., 2018), a technique used for novelty detection in reinforcement learning. PEC can be viewed as an approximation of a classification rule based on Gaussian Process posterior variance (Ciosek et al., 2020), which makes it well-motivated and connected to existing theory. PEC also offers several important practical advantages as a method for CIL. Firstly, it is highly sample efficient, as demonstrated by its strong performance with a limited number of iterations. Secondly, it is a simple and easy-to-tune approach that does not introduce any extra hyperparameters. Thirdly, the absence of forgetting is guaranteed by the separation of modules for different classes. Lastly, PEC can operate even when the data are presented only one class at a time.

Since PEC was designed with efficiency in mind, we focus in our evaluations on the challenging single-pass-through-data, or online, setting (Chaudhry et al., 2019a;b; Aljundi et al., 2019a; Soutif-Cormerais et al., 2023). This setup emphasizes being able to learn from a stream of data, with limited samples, memory, and compute (Chaudhry et al., 2019a). We conduct comprehensive experiments on CIL sequences utilizing the MNIST, SVHN, CIFAR-10, CIFAR-100, and miniImageNet datasets.

Our contributions include:

- We introduce Prediction Error-based Classification (PEC), a novel approach for class-incremental learning. PEC is motivated as an efficient alternative for generative classification, and can additionally be interpreted as an approximation of a rule based on Gaussian Process posterior variance.
- We demonstrate the strong performance of PEC in single-pass-through-data CIL. Our method outperforms other rehearsal-free baselines in all cases, and rehearsal-based approaches with moderate replay buffer size in most cases.
- We provide additional experiments analyzing the crucial factors for PEC's strong performance.

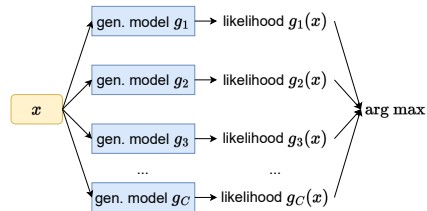

(a) In *discriminative classification*, an example $x$ is processed by a common classifier network, and the class with the largest logit is chosen.

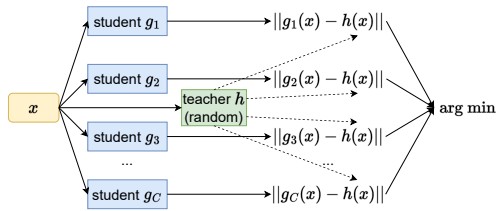

(b) In *generative classification*, the likelihood of $x$ is evaluated under each class-specific generative model, and the class with the highest one is chosen.

(c) In *Prediction Error-based Classification*, $x$ is passed through class-specific student networks and a shared teacher network, which is fixed with randomly initialized weights. The class whose student approximates the teacher most closely is chosen.

Figure 1: Comparison of discriminative classification, generative classification, and Prediction Error-based Classification.

## 2 PRELIMINARIES

### 2.1 CLASS-INCREMENTAL LEARNING

Continual learning is concerned with training machine learning models in an incremental manner, where the training data are not i.i.d. but their distribution can shift during the learning process (De Lange et al., 2022; Parisi et al., 2019). Depending on assumptions such as the type of non-stationarity and access to task identity, different types of continual learning have been distinguished (van de Ven et al., 2022). Arguably, among the most challenging of these is class-incremental learning (CIL) (Masana et al., 2023; Zhou et al., 2023; van de Ven et al., 2022).

For simplicity as well as to be comparable to previous work, in our evaluations we focus on *task-based* class-incremental learning, where the learning process is divided into clearly separated tasks that do not repeat (Aljundi et al., 2019b; van de Ven et al., 2021). However, it is important to note that PEC is not limited to this setting, as it does not make use of this task structure. In task-based CIL, an algorithm is presented with a sequence of classification tasks $T_1, T_2, \ldots, T_\tau$. During each task $T_i$, a dataset $D^i := \{(x_j^i, y_j^i)\}_j$ is presented, where each $y_j^i$ belongs to a set of possible classes $\mathcal{C}^{T_i}$. At test time, the algorithm must decide to which class a given input $x$ belongs, among all possible classes $\bigcup_{i \in \{1,2,\ldots,\tau\}} \mathcal{C}^{T_i}$. Task identity is not provided during testing, meaning that the algorithm must discriminate not only between classes that have been seen together but also between those that have not.

## 2.2 Approaches to class-incremental learning

Most approaches to CIL rely on discriminative classification (see Figure 1a) together with some mechanism to prevent forgetting. Among those, some methods use a memory buffer in which data from past tasks are stored, while others refrain from it. As storing data might be forbidden in some protocols, it is often considered a desirable property for CIL methods to not rely on a memory buffer.

When stored data from past tasks are available, the typical approach is to revisit that data when training on new data. For instance, **Experience Replay (ER)** (Chaudhry et al., 2019b) directly retrains on the buffer data, while **A-GEM** (Chaudhry et al., 2019a) uses the stored data for gradient projections to reduce interference between tasks. **Dark Experience Replay (DER)** (Buzzega et al., 2020) is a form of ER that combines replay with self-distillation. Further variants of DER include **DER++** and **X-DER** (Boschini et al., 2022). **ER-ACE** (Caccia et al., 2021) modifies the loss to only affect the scores corresponding to current classes on non-rehearsal data. **iCaRL** (Rebuffi et al., 2017) combines replay with a nearest-mean-of-exemplars classifier in the feature space. **BiC** (Wu et al., 2019) additionally uses stored data to calibrate logits for classes coming from different tasks, and thus can be classified as a *bias-correcting algorithm*.

Among methods that do not store data, sometimes referred to as *rehearsal-free*, there are *regularization-based* approaches that add an auxiliary term to the loss function, discouraging large changes to parts of the model important for past tasks. **EWC** (Kirkpatrick et al., 2017) relies on a diagonal approximation of the Fisher matrix to weigh the contributions from different parameters. **SI** (Zenke et al., 2017) maintains and updates per-parameter importance measures in an online manner. **LwF** (Li & Hoiem, 2017) uses a distillation loss to keep predictions consistent with the ones from a historical model. Additionally, there are rehearsal-free bias-correcting algorithms such as the **Labels trick** (Zeno et al., 2018), which only incorporates classes from the current task in the loss function to prevent negative bias on past tasks. Since the Labels trick often improves the performance when added on top of rehearsal-free methods, we also consider **EWC + LT**, **SI + LT**, **LwF + LT** variants.

Another paradigm for CIL, which also does not require a memory buffer, is generative classification, where class-conditional generative models are leveraged directly for classification, see Figure 1b. For example, in an approach described in van de Ven et al. (2021), a separate variational autoencoder (VAE) (Kingma & Welling, 2013) is trained for every class; we refer to this method as **VAE-GC**. However, when only a single pass through the data is allowed, VAE models might be too complex, and generative classifiers based on simpler models might be preferred. One option is **SLDA** (Hayes & Kanan, 2020), which can be interpreted as learning a Gaussian distribution for each class with a shared covariance matrix (van de Ven et al., 2021). This can be simplified further by using an identity covariance matrix, in which case it becomes the **Nearest mean** classifier.

# 3 Method

## 3.1 PEC algorithm

In this section, we describe Prediction Error-based Classification (PEC), which is schematically illustrated in Figure 1c. Inspired by Random Network Distillation (Burda et al., 2018), our approach entails creating a distinct *student* neural network $g_{\theta^c}$ for each class $c$, which is trained to replicate the outputs of a frozen random *teacher* network $h_\phi : \mathbb{R}^n \to \mathbb{R}^d$ for data from that class. Here, $n$ and $d$ represent the input and output dimensions, respectively, and the output dimension can be chosen arbitrarily. The intuition is that after training, $g_{\theta^c}$ will be competent at mimicking $h_\phi$ for the inputs $x$ that are either from the training inputs set $X_c$, or close to the examples from $X_c$. We leverage here the generalization property of neural networks (Kawaguchi et al., 2017; Goodfellow et al., 2016), which tend to exhibit low error not only on training data but also on unseen test data from the same distribution. If, on the other hand, some out-of-distribution input $\widetilde{x}$ is considered, then the error $||g_{\theta^c}(\widetilde{x}) - h_\phi(\widetilde{x})||$ is unlikely to be small (Ciosek et al., 2020), as the network was not trained in that region. To classify using PEC, we feed the input $x$ through all networks $g_{\theta^c}$ corresponding to different classes and select the class with the smallest error $||g_{\theta^c}(x) - h_\phi(x)||$. The training and inference procedures are outlined in Algorithm 1 and Algorithm 2, respectively.

As the architectures for $g$ and $h$, we use shallow neural networks with one hidden layer, either convolutional or linear, followed by respectively a ReLU or GELU nonlinearity, and then a final linear layer. Additionally, the teacher network's middle layer is typically wider than the student's one. A detailed description of the used architectures is deferred to Appendix B.1. Note that the student networks from different classes can be implemented as a single module in an automatic differentiation framework, jointly computing all class-specific outputs. This means that during inference only two forward passes are needed to obtain the classification scores – one for such a merged student, and one for the teacher network. We use the Kaiming initialization (He et al., 2015), the default in PyTorch,

---

**Algorithm 1** Training of PEC

---

**input:** Datasets $X_c$ and student networks $g_{\theta^c}$ for classes $c \in \{1, 2, \ldots, C\}$, teacher network $h_\phi$, number of optimization steps $S$
**output:** Trained student networks $g_{\theta^c}$, randomly intialized teacher network $h_\phi$
Initialize $\theta^1, \theta^2, \ldots, \theta^C$ with the same random values
Initialize $\phi$ with random values
**for** class $c \in \{1, 2, \ldots, C\}$ **do**
    **for** iteration $i \in \{1, 2, \ldots, S\}$ **do**
        $x \leftarrow$ batch of examples from $X_c$
        Update $\theta^c$ to minimize $||g_{\theta^c}(x) - h_\phi(x)||^2$
    **end for**
**end for**

---

---

**Algorithm 2** Inference using PEC

---

**input:** Example $x$, student networks $g_{\theta^c}$ for classes $c \in \{1, 2, \ldots, C\}$, teacher network $h_\phi$
**output:** Predicted class $c_p$
**return** $c_p := \arg\min_{c \in \{1, 2, \ldots, C\}} ||g_{\theta^c}(x) - h_\phi(x)||$

---

to initialize both the teacher and student networks. We ablate this choice in Appendix C.7 and find that PEC's performance is relatively stable with respect to the initialization scheme.

This classification paradigm is well-suited for class-incremental learning since each class has its own dedicated model, eliminating interference and preventing forgetting. Furthermore, the performance of PEC does not depend on task boundaries (e.g., blurry task boundaries are no problem) and the approach works well even if each task contains only one class. This is unlike most discriminative classification methods, which struggle in such a setting since they operate by contrasting classes against one another, in consequence requiring more than one class. Moreover, as PEC works well with a batch size of 1, it is also well suited for streaming settings. Additionally, PEC has the advantage of not introducing any new hyperparameters beyond architecture and learning rate, which are also present in all considered baselines.

PEC can be motivated from two different angles. One perspective is that PEC is related to the generative classification paradigm, but it replaces the difficult generative task with a much simpler, random task. As will be shown in the experimental section, this leads to significant efficiency improvements. Alternatively, PEC can be seen as an approximation of a principled classification rule based on the posterior variance of a Gaussian Process. We delve into this perspective in the following section.

### 3.2 THEORETICAL SUPPORT FOR PEC

In this section, we first explain the classification rule based on Gaussian Process posterior variance, and then we describe how this rule is approximated by our PEC.

**Gaussian Processes** A Gaussian Process (GP) is a stochastic process, i.e. a collection of random variables $\{H(x)\}$ indexed by values $x \in \mathbb{R}^n$ for some $n$, such that every finite sub-collection of the random variables follows a Gaussian distribution (Rasmussen & Williams, 2006). Typically, a GP is defined in terms of a mean function $\mu : \mathbb{R}^n \to \mathbb{R}$ and a covariance function $k : \mathbb{R}^n \times \mathbb{R}^n \to \mathbb{R}$. Given some dataset $(X, y)$, a given prior GP $H$ can be updated to a posterior GP $H_{(X,y)}$. Let us denote by $\sigma(H_X)(x^*)$ the posterior variance of $H_{(X,y)}$ at point $x^*$. A useful property of $\sigma(H_X)(x^*)$ is that it depends only on the inputs $X$ and not on the targets $y$ (Rasmussen & Williams, 2006). The posterior variance of a GP is often interpreted as a measure of uncertainty (Ciosek et al., 2020).

**Classification rule based on GP posterior variance** Let $H$ be a Gaussian Process. Let us assume distributions $\mathcal{X}_1, \ldots, \mathcal{X}_C$ for the data from $C$ different classes, and finite datasets $X_1 \sim \mathcal{X}_1, \ldots, X_C \sim \mathcal{X}_C$. Given some test data point $x^*$, consider the following classification rule:

$$\text{class}(x^*) := \arg\min_{1 \le c \le C} \sigma(H_{X_c})(x^*). \tag{1}$$

Let us informally motivate this rule. Assume that $x^*$ belongs to the class $\bar{c}$, for some $1 \le \bar{c} \le C$. Then, in the dataset $X_{\bar{c}}$ for the class $\bar{c}$, there are many examples close to $x^*$, which in consequence make the posterior variance $\sigma(H_{X_{\bar{c}}})(x^*)$ close to 0. In contrast, for any $c \ne \bar{c}$, examples from $X_c$ are more distant from $x^*$, which means that the reduction from the prior variance $\sigma(H)(x^*)$ to the posterior variance $\sigma(H_{X_c})(x^*)$ is smaller. Hence, it is intuitively clear that the rule in Equation 1 should typically choose the correct class $\bar{c}$. In Appendix D, we provide formal support for this argument.

**PEC approximates GP posterior variance classification rule** Here, we argue that PEC approximates the GP posterior variance classification rule, using tools from Ciosek et al. (2020). For a given GP $H$, a dataset $X$, and a data point $x^*$, define:

$$s_B^{H,X}(x^*) := \frac{1}{B} \sum_{i=1}^{B} ||g^{h_i,X}(x^*) - h_i(x^*)||^2, h_i \sim H; \tag{2}$$

$$s_\infty^{H,X}(x^*) := \mathbb{E}_{h \sim H} ||g^{h,X}(x^*) - h(x^*)||^2, \tag{3}$$

where $B$ is the number of samples from GP $H$ and $g^{h,X}$ is a neural network trained to imitate $h$ on dataset $X$, using some fixed procedure. That is, $s_B^{H,X}(x^*)$, for $B \in \mathbb{Z}_+ \cup \{\infty\}$, measures the average prediction error at $x^*$ of a neural network trained with dataset $X$ to approximate a function sampled from $H$. Now, let us introduce GP-PEC($B$), a modified version of PEC that we will use in our analyses. Given classes $1 \leq c \leq C$ with datasets $X_1, \ldots, X_C$, GP-PEC($B$), for $B \in \mathbb{Z}_+ \cup \{\infty\}$, classifies using the following rule:

$$\text{class}(x^*) := \underset{1 \leq c \leq C}{\arg\min} \, s_B^{H,X_c}(x^*). \tag{4}$$

We can connect PEC to GP-PEC($B$) by using a sample size of 1 and by replacing the GP "teacher" $H$ with a neural network teacher $h$. Note that our theory does not quantify errors arising from this replacement; however, the replacement is justified since wide neural networks with random initializations approximate Gaussian Processes (Neal, 1996; Yang, 2019). In our experiments, we use a wide hidden layer for the teacher network, a choice that we ablate in Section 4.5.

Next, we introduce propositions connecting GP-PEC($B$) with the GP posterior variance classification rule. The proofs for these propositions are provided in Appendix D.

**Proposition 1** (Proposition 1 from Ciosek et al. (2020), rephrased). *Let $H$ be a Gaussian Process and $X$ a dataset. Then the following inequality holds:*

$$s_\infty^{H,X}(x^*) \geq \sigma(H_X)(x^*). \tag{5}$$

**Proposition 2.** *Let $H$ be a Gaussian Process, $\mathcal{X}$ be a data distribution, and $X := \{x_1, \ldots, x_N\} \sim \mathcal{X}$ a finite dataset. Assume that our class of approximators is such that for any $h \sim H$, $h$ can be approximated by $g^{h,X}$ so that, for arbitrarily large training sets, the training error is smaller than $\epsilon > 0$ with probability at least $1 - \gamma$, $\gamma > 0$. Then, under technical assumptions, for any $\delta > 0$, it holds with probability at least $1 - \delta - \gamma$ that*

$$\mathbb{E}_{x^* \sim \mathcal{X}} \left[ s_1^{H,X}(x^*) \right] \leq \epsilon + \kappa_N, \tag{6}$$

*where $\kappa_N$ goes to 0 as $N \to \infty$.*

Proposition 1 states that the scores used by GP-PEC($\infty$) overestimate scores from the GP posterior variance rule. On the other hand, Proposition 2 shows that on average, the score assigned by GP-PEC(1) for the correct class will go to 0 as $N \to \infty$, given appropriately strong approximators and technical assumptions listed in Appendix D. Together, these propositions indicate that GP-PEC($B$), and by extension PEC, approximate the GP posterior variance classification rule in Equation 1.

## 4 EXPERIMENTS

We present an experimental evaluation of PEC on several class-incremental learning benchmarks. Our main result is that PEC performs strongly in single-pass-through-data CIL, outperforming other rehearsal-free methods in all cases and rehearsal-based methods with moderate replay buffer size in most cases. This section is structured as follows: first, we explain the experimental setup in Section 4.1. Then, we present the benchmarking results, with Section 4.2 focusing on the single-pass-through-data setting, and Section 4.3 covering a broader spectrum of regimes. In Section 4.4, we examine the comparability of PEC scores for different classes. Finally, Section 4.5 investigates the impact of various architecture choices on PEC's performance.

### 4.1 EXPERIMENTAL SETUP

All experiments follow the class-incremental learning scenario, and, unless otherwise noted, for each benchmark we only allow a single pass through the data, i.e. each training sample is seen only once except if it is stored in the replay buffer. Performance is evaluated as the final test accuracy after training on all classes. We adopt Adam (Kingma & Ba, 2014) as the optimizer. We provide additional details about the experimental setup in Appendix B.

Table 1: Empirical evaluation of PEC against a suite of CIL baselines in case of one class per task. The single-pass-through-data setting is used. Final average accuracy (in %) is reported, with mean and standard error from 10 seeds. Parameter counts for all methods are matched, except for those indicated with [†], see Section 4.1.

| | Method | Buffer | MNIST (10/1) | Balanced SVHN (10/1) | CIFAR-10 (10/1) | CIFAR-100 (100/1) | miniImageNet (100/1) |
|---|---|---|---|---|---|---|---|
| | *Fine-tuning* | | 10.09 ($\pm$ 0.00) | 10.00 ($\pm$ 0.00) | 9.99 ($\pm$ 0.01) | 1.09 ($\pm$ 0.04) | 1.02 ($\pm$ 0.02) |
| | *Joint, 1 epoch* | | 97.16 ($\pm$ 0.04) | 92.69 ($\pm$ 0.08) | 74.16 ($\pm$ 0.13) | 34.87 ($\pm$ 0.17) | 25.29 ($\pm$ 0.15) |
| | *Joint, 50 epochs* | | 98.26 ($\pm$ 0.03) | 95.88 ($\pm$ 0.03) | 93.30 ($\pm$ 0.08) | 73.68 ($\pm$ 0.08) | 72.26 ($\pm$ 0.12) |
| discriminative | ER | ✓ | 84.42 ($\pm$ 0.26) | 74.80 ($\pm$ 0.69) | 40.59 ($\pm$ 1.17) | 5.14 ($\pm$ 0.21) | 5.72 ($\pm$ 0.24) |
| | A-GEM | ✓ | 59.84 ($\pm$ 0.82) | 9.93 ($\pm$ 0.03) | 10.19 ($\pm$ 0.11) | 1.05 ($\pm$ 0.03) | 1.11 ($\pm$ 0.07) |
| | DER | ✓ | 91.65 ($\pm$ 0.11) | 76.45 ($\pm$ 0.71) | 40.03 ($\pm$ 1.52) | 1.05 ($\pm$ 0.07) | 0.99 ($\pm$ 0.01) |
| | DER++ | ✓ | 91.88 ($\pm$ 0.19) | **80.66 ($\pm$ 0.48)** | 35.61 ($\pm$ 2.43) | 6.00 ($\pm$ 0.25) | 1.40 ($\pm$ 0.09) |
| | X-DER | ✓ | 82.97 ($\pm$ 0.14) | 70.65 ($\pm$ 0.61) | 43.16 ($\pm$ 0.46) | 15.57 ($\pm$ 0.27) | 8.22 ($\pm$ 0.35) |
| | ER-ACE | ✓ | 87.78 ($\pm$ 0.19) | 74.47 ($\pm$ 0.33) | 39.90 ($\pm$ 0.46) | 8.21 ($\pm$ 0.21) | 5.73 ($\pm$ 0.17) |
| | iCaRL | ✓ | 83.08 ($\pm$ 0.28) | 57.18 ($\pm$ 0.94) | 37.80 ($\pm$ 0.42) | 8.61 ($\pm$ 0.13) | 7.52 ($\pm$ 0.14) |
| | BiC | ✓ | 86.00 ($\pm$ 0.37) | 70.69 ($\pm$ 0.52) | 35.86 ($\pm$ 0.38) | 4.63 ($\pm$ 0.20) | 1.46 ($\pm$ 0.05) |
| | Labels trick (LT) | - | 10.93 ($\pm$ 0.87) | 9.93 ($\pm$ 0.08) | 10.00 ($\pm$ 0.22) | 1.04 ($\pm$ 0.05) | 0.97 ($\pm$ 0.04) |
| | EWC + LT | - | 9.20 ($\pm$ 0.91) | 9.90 ($\pm$ 0.09) | 10.54 ($\pm$ 0.17) | 1.01 ($\pm$ 0.05) | 1.03 ($\pm$ 0.03) |
| | SI + LT | - | 10.75 ($\pm$ 1.03) | 10.01 ($\pm$ 0.06) | 10.01 ($\pm$ 0.26) | 0.93 ($\pm$ 0.05) | 0.94 ($\pm$ 0.04) |
| | LwF + LT | - | 9.35 ($\pm$ 0.72) | 9.99 ($\pm$ 0.07) | 9.57 ($\pm$ 0.52) | 1.00 ($\pm$ 0.09) | 0.97 ($\pm$ 0.02) |
| generative | Nearest mean[†] | - | 82.03 ($\pm$ 0.00) | 12.46 ($\pm$ 0.00) | 27.70 ($\pm$ 0.00) | 9.97 ($\pm$ 0.00) | 7.53 ($\pm$ 0.00) |
| | SLDA[†] | - | 88.01 ($\pm$ 0.00) | 21.30 ($\pm$ 0.00) | 41.37 ($\pm$ 0.01) | 18.83 ($\pm$ 0.01) | 12.87 ($\pm$ 0.01) |
| | VAE-GC | - | 83.98 ($\pm$ 0.47) | 50.71 ($\pm$ 0.62) | 42.71 ($\pm$ 1.28) | 19.70 ($\pm$ 0.12) | 12.10 ($\pm$ 0.07) |
| | **PEC (ours)** | - | **92.31 ($\pm$ 0.13)** | 68.70 ($\pm$ 0.16) | **58.94 ($\pm$ 0.09)** | **26.54 ($\pm$ 0.11)** | **14.90 ($\pm$ 0.08)** |

**Tasks and datasets** To construct sequences of tasks, we split up popular image classification datasets such that each task contains data from a subset of classes and those subsets are disjoint. Following Masana et al. (2023), if a dataset $D$ is split into $T$ tasks with $C$ classes each, the resulting sequence is denoted $D(T/C)$. For each dataset, we consider two splits: a standard one, with multiple classes in each task, and a more challenging one, with only one class per task. For **MNIST** (Deng, 2012), **Balanced SVHN**[2] (Netzer et al., 2011), and **CIFAR-10** (Krizhevsky et al., 2009), we use $(5/2)$ and $(10/1)$ splits. For **CIFAR-100** (Krizhevsky et al., 2009), we use $(10/10)$ and $(100/1)$ splits. For **miniImageNet** (Vinyals et al., 2016), we use $(20/5)$ and $(100/1)$ splits.

**Methods** We compare PEC against a comprehensive list of CIL baselines, see Section 2.2. Additionally, we include **Fine-tuning**, which trains a discriminative classifier on a sequence of tasks without employing any continual learning strategy, and **Joint**, which can be seen as an upper target that jointly trains on all the classes at the same time. For the non-rehearsal approaches, we report the variants with added Labels trick in the main text; we provide vanilla results in Appendix C.2.

**Architectures** Total parameter counts used by models for each method are matched, except for Nearest mean and SLDA[3]. For methods based on discriminative classification, we use standard architectures: for MNIST, an MLP with two hidden layers, 100 neurons each; for other datasets, a variant of ResNet18 (He et al., 2016) with the initial downscaling removed, as in Buzzega et al. (2020). For VAE-GC and PEC, which maintain separate modules per class, we use small and shallow neural networks to match their total parameter count with other approaches; in particular, we do not use ResNets for those methods. We describe details in Appendix B.1.

**Replay buffers** Although PEC does not use a replay buffer, we also compare against rehearsal-based baselines. We grant these a replay buffer of 500 examples (Buzzega et al., 2020; Boschini et al., 2022). This is additional memory that these methods can use; the rehearsal-free methods are not compensated for this. For each batch of data from the current task containing $b$ examples, a rehearsal-based method can access the replay buffer and use $b$ extra examples from it.

**Hyperparameters** For each combination of method, dataset, and task split, we perform a separate grid search to select the learning rate, the batch size, whether to use data augmentation, whether to use linear learning rate decay, and method-dependent hyperparameters. Note that PEC does not introduce the latter. Details on the selected values and search grids are deferred to Appendix B.3.

---

[2] For the main experiments (Tables 1 and 2), we use Balanced SVHN, a version of SVHN with equalized numbers of samples per class. We do this to isolate the issue of data imbalance, discussed separately in Section 4.4.

[3] As these methods rely on the estimation of data means and covariances, it is not directly possible to manipulate the number of parameters used. We report numbers of parameters for all the methods in Appendix B.2.

Table 2: Empirical evaluation of PEC against a suite of CIL baselines in case of multiple classes per task. The single-pass-through-data setting is used. Final average accuracy (in %) is reported, with mean and standard error from 10 seeds. For joint training and generative classifiers, results are the same as with one class per task (see Table 1) and omitted. Results for PEC are also the same, but included to ease comparison. Appendix B.2 explains why these results are the same. Parameter counts for all methods are matched.

| | Method | Buffer | MNIST (5/2) | Balanced SVHN (5/2) | CIFAR-10 (5/2) | CIFAR-100 (10/10) | miniImageNet (20/5) |
|---|---|---|---|---|---|---|---|
| | *Fine-tuning* | | 19.57 ($\pm$ 0.01) | 19.14 ($\pm$ 0.03) | 18.54 ($\pm$ 0.17) | 6.25 ($\pm$ 0.08) | 3.51 ($\pm$ 0.04) |
| discriminative | ER | ✓ | 86.55 ($\pm$ 0.17) | 80.00 ($\pm$ 0.54) | 57.49 ($\pm$ 0.54) | 11.41 ($\pm$ 0.22) | 6.67 ($\pm$ 0.21) |
| | A-GEM | ✓ | 50.82 ($\pm$ 1.36) | 19.71 ($\pm$ 0.49) | 18.21 ($\pm$ 0.14) | 5.25 ($\pm$ 0.14) | 2.47 ($\pm$ 0.18) |
| | DER | ✓ | 91.64 ($\pm$ 0.11) | 78.22 ($\pm$ 0.94) | 43.98 ($\pm$ 1.20) | 7.39 ($\pm$ 0.06) | 3.86 ($\pm$ 0.04) |
| | DER++ | ✓ | **92.30 ($\pm$ 0.14)** | **86.50 ($\pm$ 0.43)** | 50.61 ($\pm$ 0.80) | 19.99 ($\pm$ 0.28) | 10.50 ($\pm$ 0.42) |
| | X-DER | ✓ | 81.91 ($\pm$ 0.53) | 73.88 ($\pm$ 1.59) | 54.15 ($\pm$ 1.92) | **28.64 ($\pm$ 0.22)** | **14.65 ($\pm$ 0.40)** |
| | ER-ACE | ✓ | 89.97 ($\pm$ 0.29) | 83.30 ($\pm$ 0.33) | 53.25 ($\pm$ 0.69) | 21.02 ($\pm$ 0.32) | 8.95 ($\pm$ 0.35) |
| | iCaRL | ✓ | 80.41 ($\pm$ 0.45) | 73.65 ($\pm$ 0.57) | 54.33 ($\pm$ 0.33) | 21.60 ($\pm$ 0.20) | 13.46 ($\pm$ 0.20) |
| | BiC | ✓ | 88.08 ($\pm$ 0.81) | 77.65 ($\pm$ 0.45) | 46.74 ($\pm$ 0.83) | 13.16 ($\pm$ 0.28) | 6.90 ($\pm$ 0.19) |
| | Labels trick (LT) | - | 38.34 ($\pm$ 2.46) | 31.86 ($\pm$ 0.81) | 26.19 ($\pm$ 0.38) | 7.84 ($\pm$ 0.15) | 4.44 ($\pm$ 0.08) |
| | EWC + LT | - | 46.41 ($\pm$ 1.04) | 26.04 ($\pm$ 1.56) | 26.51 ($\pm$ 1.10) | 8.64 ($\pm$ 0.26) | 3.64 ($\pm$ 0.16) |
| | SI + LT | - | 52.45 ($\pm$ 1.84) | 30.81 ($\pm$ 0.86) | 25.80 ($\pm$ 0.78) | 8.45 ($\pm$ 0.14) | 4.76 ($\pm$ 0.06) |
| | LwF + LT | - | 55.38 ($\pm$ 1.27) | 52.21 ($\pm$ 1.22) | 40.40 ($\pm$ 0.91) | 14.38 ($\pm$ 0.17) | 3.74 ($\pm$ 0.33) |
| | **PEC (ours)** | - | **92.31 ($\pm$ 0.13)** | 68.70 ($\pm$ 0.16) | **58.94 ($\pm$ 0.09)** | 26.54 ($\pm$ 0.11) | **14.90 ($\pm$ 0.08)** |

## 4.2 PERFORMANCE OF PEC

Here we present the main benchmarking results. Evaluations for the cases of one class per task and multiple classes per task are presented in Table 1 and Table 2, respectively. PEC performs strongly in both cases, achieving the highest score in 7 out of 10 considered settings.

The performance of PEC is especially good in the challenging case of one class per task, see Table 1. Discriminative classification-based approaches struggle in this setting, whereas PEC does not rely on the task structure and achieves the same results for both task splits. On most datasets, PEC substantially outperforms even recently proposed replay-based methods with buffers of 500 examples.

When only comparing against rehearsal-free methods, PEC achieves the strongest performance in all considered settings. In particular, we see that PEC outperforms VAE-GC, a strong rehearsal-free baseline, as well as the other included generative classifiers. Below we also report that PEC reaches a given performance in a smaller number of iterations and with less model parameters compared to VAE-GC, see Figure 2. This fulfills our original goal to develop a method inspired by generative classification but more efficient. We provide additional experiments in Appendix C.1, where we use the experimental setups from van de Ven et al. (2021), including multi-epoch learning and parameter counts more suitable for VAE-GC, as well as the usage of pre-trained feature extractors in some cases. We show that PEC prevails also in those setups, which further corroborates its robust performance.

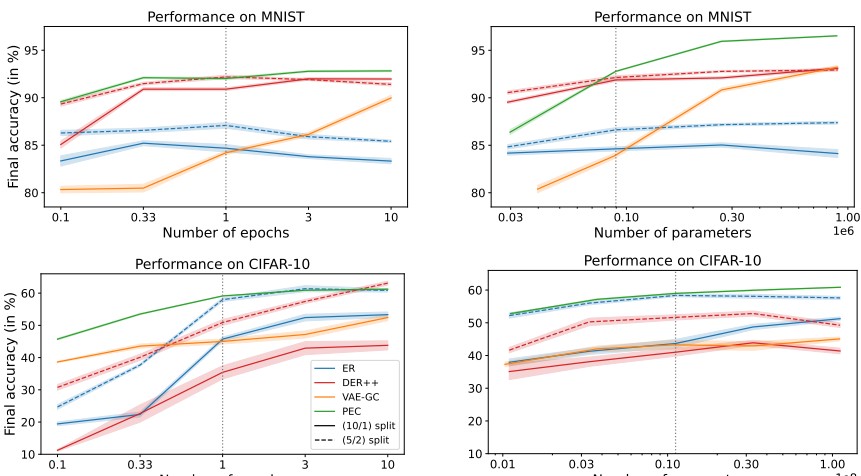

Figure 2: CIL performance for varying number of epochs (left) and number of model parameters (right). Plotted are the means (solid or dashed lines) and standard errors (shaded areas) from 10 seeds. For PEC and VAE-GC, results are the same for both task splits (see Appendix B.2), and hence single curves are shown. Vertical dotted lines indicate the settings used for the experiments in Tables 1 and 2.

## 4.3 PERFORMANCE IN VARYING REGIMES

In this section, we present a comparison of the performance of a representative set of approaches in a wider range of experimental conditions. Specifically, we evaluate ER, DER++, VAE-GC, and PEC while varying two factors: the number of epochs (Figure 2, left) and the number of model parameters (Figure 2, right). We keep the rest of the experimental setup the same as before. We use the hyperparameters as in Section 4.2, except for the learning rates, which we tune for every setting separately.

We observe good performance of PEC across the board, confirming its robustness to varying experimental conditions. PEC outperforms ER and VAE-GC in all considered cases, and DER++ in most cases. Increasing the number of epochs generally benefits all methods, especially for CIFAR-10. Conversely, scaling the number of parameters beyond the levels used in the main experiments improves the performance of PEC and VAE-GC, but less so for the discriminative approaches, ER and DER++.

## 4.4 COMPARABILITY OF PEC CLASS SCORES

In this section, we investigate whether the magnitudes of the scores computed by PEC for different classes are comparable. In Section 3.2, we provided support for the use of PEC scores in the limit of data and ensemble members. However, in practice, we use a limited dataset and a single student-teacher pair. Thus, it is not clear *a priori* whether the obtained scores for different classes would be comparable out of the box.

Table 3: Performance of PEC under balanced and imbalanced data, as well as with different mitigation strategies. Reported is the final average accuracy (in %), with mean and standard error from 10 seeds.

| Method | CIFAR-10 | Imbalanced CIFAR-10 | SVHN |
|---|---|---|---|
| PEC | 58.94 ($\pm$ 0.09) | 28.96 ($\pm$ 0.21) | 21.73 ($\pm$ 0.14) |
| PEC, Equal budgets | 58.94 ($\pm$ 0.09) | 55.84 ($\pm$ 0.14) | 68.65 ($\pm$ 0.21) |
| PEC, Oracle balancing | 61.01 ($\pm$ 0.11) | 59.13 ($\pm$ 0.47) | 68.55 ($\pm$ 0.09) |
| PEC, Buffer balancing | 59.02 ($\pm$ 0.25) | 57.31 ($\pm$ 0.60) | 66.82 ($\pm$ 0.25) |

To address this question, we examine if the performance of PEC on the CIFAR-10 dataset can be improved by re-scaling the obtained class scores using per-class scalars. For this purpose, we do the following. After the training of PEC, we use the CMA-ES (Hansen & Ostermeier, 2001) evolutionary optimization algorithm (see Appendix B.4 for details) to find a scalar coefficient $s_c$ for each class $c$ to scale its class score; then, $s_c||g_{\theta^c}(x) - h_\phi(x)||^2$ is used as a score for class $c$. We consider two procedures. **Oracle balancing** uses the full test set to find the optimal $s_c$ coefficients, and **Buffer balancing** uses the replay buffer of 500 samples from the training set to do so. We observe in Table 3 that Oracle balancing is able to improve upon the baseline version of PEC, although not by a wide margin, while Buffer balancing does not bring a significant improvement. This suggests good out-of-the-box comparability of the PEC class scores for a balanced dataset like CIFAR-10.

However, in initial experiments we found that an imbalanced training dataset, where classes have different numbers of samples, can severely impair the performance of our method. To study the impact of dataset imbalance, we compare the performance of PEC on CIFAR-10 and Imbalanced CIFAR-10. The latter is a modified version of CIFAR-10, where the number of examples per class is doubled for class 0 and halved for class 1, causing dataset imbalance; the test dataset stays the same. We also study SVHN, which is imbalanced in its original form; in Section 4.2, we reported results for the balanced variant. We observe that in the case of imbalanced datasets, the performance of vanilla PEC drops substantially compared to balanced counterparts, see Table 3. At the same time, we find that this deterioration is mostly mitigated by the balancing strategies that were introduced above, which suggests that dataset imbalance affects the comparability of the PEC class scores.

Note that Buffer balancing is a valid strategy that could be used in practice if the replay buffer is allowed. For the setting without buffers, we additionally propose **Equal budgets**, which is a simple strategy that equalizes the numbers of training iterations for different classes, see Appendix B.4 for details. We observe that Equal budgets also effectively mitigates the effect of dataset imbalance, see Table 3.

## 4.5 IMPACT OF ARCHITECTURAL CHOICES

In this section, we examine various architectural choices for PEC and their impact on performance. Figure 3 shows our analysis of the following factors: number of hidden layers (for both the student and the teacher network), width of the hidden layer of the student network, width of the hidden layer of the teacher network, and output dimension. For every considered setting, the learning rate is tuned.

We observe that increased depth does not seem to enhance PEC's performance. On the other hand, scaling up the other considered variables leads to consistent improvements for both MNIST and CIFAR-10; note that these correspond to different ways of scaling width. We see that the output dimensions and the width of the teacher's hidden layer seem particularly important. The impact of the latter can be understood through the lens of the GP-based classification rule, where the teacher is

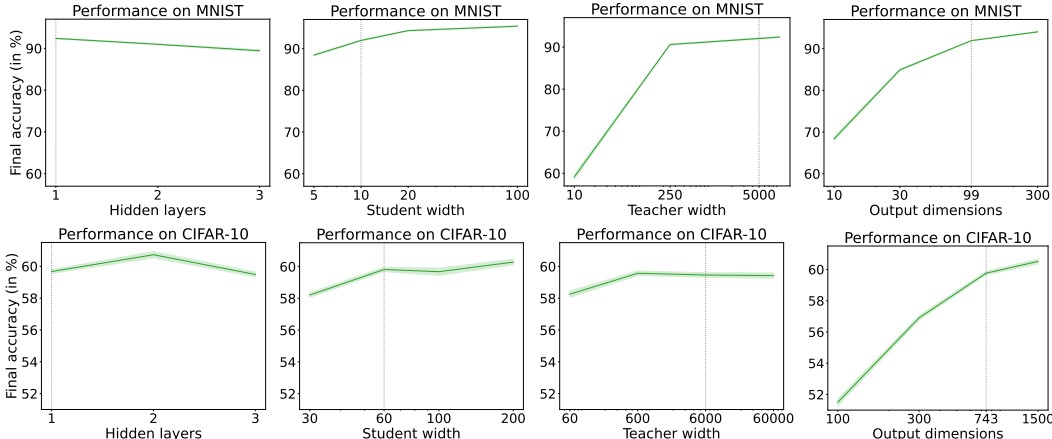

Figure 3: Impact of architectural choices on PEC's performance. Plotted are the means (solid lines) and standard errors (shaded areas) from 10 seeds. Vertical dotted lines mark settings used for experiments in Tables 1 and 2.

treated as approximating a sample from a Gaussian Process, see Section 3.2; it is known that neural networks approximate Gaussian Processes as their width increases (Neal, 1996).

## 5 RELATED WORK

We discuss relevant literature on continual learning and class-incremental learning in Section 2.

**Random Network Distillation**  An important foundational work to our study is Burda et al. (2018), where Random Network Distillation is proposed. This method consists of cloning a random frozen neural network into a trained one and then utilizing prediction errors to detect novel states in a reinforcement learning environment. While we reuse this principle in our PEC, we note that to the best of our knowledge, this approach has not previously been used for class-incremental learning, or even classification in general. Another work on which we build is Ciosek et al. (2020), which reinterprets Random Network Distillation as a principled way to perform uncertainty estimation for neural networks; we use ideas from this work to build our theoretical support for PEC.

**OOD detection for task inference**  The works of Kim et al. (2022a;b) decompose CIL into task-incremental learning and task inference, and then tackle the latter with out-of-distribution (OOD) detection methods. Our PEC scores could be interpreted through a similar lens. However, the approaches used in the mentioned works are not suitable in cases of single-pass-through-data or one-class-per-task. Another related work is Henning et al. (2021), which uses uncertainty-based OOD detection for task recognition.

## 6 LIMITATIONS AND FUTURE WORK

Although we have shown the effectiveness of PEC, we are also mindful of some of its limitations:

- The current version of PEC completely separates information coming from different classes. While this eliminates forgetting, it also prevents the transfer of information.
- In its current form, PEC uses a separate student model for each class, resulting in a linear increase in the number of parameters with the number of classes. However, as PEC uses small and shallow networks, the severity of this issue is limited in practice.
- As indicated in Section 4.4, the performance of PEC drops in the case of data imbalance. However, this deterioration is mostly eliminated with the proposed mitigation strategies.

An interesting, and perhaps surprising, design choice of PEC is the use of a random shallow neural network as a teacher. While this choice has been empirically justified by its strong performance, and there is theoretical support for this choice through the link with Gaussian Processes, it is possible that the performance of PEC could be improved further by considering different teacher functions. Options might be to use (pre-)trained neural networks or perceptual hashes (Zauner, 2010). We leave exploring the effectiveness of such alternative forms of the teacher function for future work.

## 7 CONCLUSIONS

In this study, we proposed Prediction Error-based Classification (PEC). This novel approach for class-incremental learning uses the prediction error of networks trained to mimic a frozen random network to generate a score for each class. We demonstrated the strong empirical performance of PEC in single-pass-through-data CIL, outperforming rehearsal-free methods in all cases and rehearsal-based methods with moderate replay buffer size in most cases.

## ACKNOWLEDGEMENTS

Michał Zając thanks Marcin Sendera and Tomasz Kuśmierczyk for helpful discussions. This paper is part of a project that has received funding from the European Union under the Horizon 2020 research and innovation program (ERC project KeepOnLearning, grant agreement No. 101021347) and under Horizon Europe (Marie Skłodowska-Curie fellowship, grant agreement No. 101067759). This research was supported by the PL-Grid Infrastructure. The computations were partially performed in the Poznań Supercomputing and Networking Center. The work of Michał Zając was supported by the National Centre of Science (Poland) Grant No. 2021/43/B/ST6/01456. Michał Zając acknowledges support from Warsaw University of Technology within the Excellence Initiative: Research University (IDUB) program.

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

Table 4: MLP architecture used in PEC experiments.

| linear hidden layer, $w$ neurons |
| --- |
| layer normalization |
| GELU activation |
| linear output layer, $d$ neurons |

Table 5: Convolutional architecture used in PEC experiments.

| convolutional hidden layer, $w$ channels, $3 \times 3$ filter, padding $= 1$ |
| --- |
| instance normalization |
| ReLU activation |
| average pooling, $r_{input} \times r_{input} \rightarrow r \times r$ |
| flattening layer |
| linear output layer, $d$ neurons |

## A  AUTHORS CONTRIBUTIONS

Michał came up with the idea for PEC, implemented all experiments, developed the theoretical argument, wrote the initial draft of the paper, contributed to the discussions, and proposed some of the experiments.

Tinne co-supervised the project, provided feedback on the experiments and the draft, contributed to the discussions, and proposed some of the experiments.

Gido supervised the project, suggested Michał to work on generative classification, provided feedback on the experiments and the draft, contributed to writing, contributed to the discussions, and proposed some of the experiments.

## B  DETAILS ON EXPERIMENTAL SETUP

We implement our experiments using PyTorch (Paszke et al., 2019) and Mammoth[4] (Buzzega et al., 2020; Boschini et al., 2022), except for the experiments in Appendix C.1, which are implemented on top of the codebase from van de Ven et al. (2021)[5]. Note that in PEC, we use default initialization from PyTorch (i.e., Kaiming initialization (He et al., 2015)) for both student and teacher networks.

We present details on architectures in Appendix B.1, on method implementations in Appendix B.2, on hyperparameters in Appendix B.3, on balancing strategies in Appendix B.4, and on data augmentations in Appendix B.5.

### B.1  ARCHITECTURES

We outline neural network architectures used in PEC experiments. MLP (used for MNIST (Deng, 2012)) is described in Table 4 and convolutional architecture (used for SVHN (Netzer et al., 2011), CIFAR-10 (Krizhevsky et al., 2009), CIFAR-100 (Krizhevsky et al., 2009), and miniImageNet (Vinyals et al., 2016)) is presented in Table 5.

Both architectures have a width parameter $w$ for the middle layer. As discussed in Section 4.5, PEC benefits from having the teacher network wider than the students, and consequently, we use separate values, $w_g$ and $w_h$, for student and teacher width, respectively. Additionally, convolutional architecture has a parameter $r$ that describes resolution after average pooling.

Below we list settings used for different datasets:

- MNIST: MLP network, $w_g = 10$, $w_h = 5000$, $d = 99$.

---

[4]GitHub repository: `https://github.com/aimagelab/mammoth`.
[5]GitHub repository: `https://github.com/GMvandeVen/class-incremental-learning`.

Table 6: Parameter counts used by the methods.

| Method | MNIST | Balanced SVHN | CIFAR-10 | CIFAR-100 | miniImageNet |
|---|---|---|---|---|---|
| Nearest mean | $7.8K$ | $30.7K$ | $30.7K$ | $307.0K$ | $2.1M$ |
| SLDA | $622.0K$ | $9.5M$ | $9.5M$ | $9.7M$ | $9.7M$ |
| All other | $89.6K$ | $11.2M$ | $11.2M$ | $11.2M$ | $11.2M$ |

Table 7: Multiply-accumulate operations (MACs) used in a single inference forward pass.

| Method | MNIST | Balanced SVHN | CIFAR-10 | CIFAR-100 | miniImageNet |
|---|---|---|---|---|---|
| PEC | $4.5M$ | $352M$ | $352M$ | $300M$ | $1.9G$ |
| discriminative | $90.0K$ | $556M$ | $556M$ | $556M$ | $3.9G$ |

- Balanced SVHN and CIFAR-10: convolutional network, $w_g = 60$, $w_h = 6000$, $d = 743$, $r = 5$.
- CIFAR-100 and miniImageNet: convolutional network, $w_g = 40$, $w_h = 4000$, $d = 172$, $r = 4$.

These values vary mostly to accommodate different parameter budgets and input sizes. In particular, values of $d$ not being round numbers is a consequence of matching a given parameter budget. In our preliminary experiments, we did some tuning of the architectures while looking at validation accuracy. However, this tuning was not performed for all datasets, and we directly reuse architectures from CIFAR-10 to Balanced SVHN, and from CIFAR-100 to miniImageNet. In the latter case, we transfer the architecture despite different input sizes, which is possible because of the use of the convolutional layer and average pooling. Moreover, given the form of the architectures from Tables 4 and 5 and some parameter budget, there is a very limited search space for the concrete architecture. Hence, we expect that the architectures we found would also perform well for other datasets with compatible input sizes.

## B.2 Details on the implementation of methods

**Parameter counts** We report total parameter counts in Table 6. For all methods except for Nearest mean and SLDA (Hayes & Kanan, 2020), these are matched. Note that for PEC, we do not count the parameters of the frozen random teacher $h$ towards the limit; since these parameters are fixed, the algorithm does not have to remember them, and can instead recreate them using only the remembered random seed.

**Number of operations needed** We compare the number of multiply-accumulate operations (MACs) needed for a single inference forward pass, comparing our PEC versus the discriminative approaches. Results in Table 7 show that PEC typically uses fewer operations. Intuitively, this is because PEC does a substantial part of its computation in a fully connected layer which has a much smaller compute-to-parameters density compared to convolutional layers that mostly constitute the ResNet architecture utilized in discriminative approaches.

**Replay-based methods** For the replay-based methods, we allow taking an extra $b$ examples from the buffer for each batch of the current data containing $b$ examples. Some of the methods (namely, DER++ (Buzzega et al., 2020) and X-DER (Boschini et al., 2022)) sample from the replay buffer multiple times for each batch of current data. We modified implementations so that the algorithm only samples once from the replay buffer, and then reuses those samples to compute the various losses defined in those methods.

**A-GEM** We use the implementation of A-GEM (Chaudhry et al., 2019a) with reservoir replay buffer from Mammoth.

**SLDA** To make this algorithm comply with the single-pass-through-data setting, we do not start with a batch-wise computation of the covariance matrix on the first task and only rely on online updates, using an identity covariance matrix as a starting point. SLDA uses *shrinkage parameter $\epsilon$*, which is a coefficient for the added identity matrix in the precision matrix computation. While originally it was proposed to be a constant equal to 0.0001, we observed in our experiments that setting much higher

values for this parameter can act as a useful regularizer and yield improved results. We describe search space and selected values for this parameter in Appendix B.3.1. Additionally, for SLDA experiments on miniImageNet, we rescale the inputs from $84 \times 84$ to $32 \times 32$ to make the covariance matrix computation feasible.

**VAE-GC** For VAE-GC we use architectures from van de Ven et al. (2021) with the widths adjusted to match parameter counts from other methods. We use 100 importance samples to estimate likelihoods. Furthermore, for VAE-GC experiments on miniImageNet, we rescale the inputs from $84 \times 84$ to $32 \times 32$ to make the generative task simpler; we found that this resolution works better than the original one.

Finally, note that for PEC and the considered generative classifiers (Nearest mean, SLDA and VAE-GC), we only report results for the one-class-per-task splits. The reason is that the performance of these methods should not depend on the task split, because the class-specific modules of these methods are only updated on examples from one class anyway. A slight subtlety is that when using a batch size larger than one, the mini-batches in the multiple-classes-per-task setting contain samples from multiple classes, thus reducing the effective per-class batch size (and making it variable, as each mini-batch does not need to contain an equal number of samples from each class in the task). For simplicity, and because for PEC we always use batch size $= 1$, we ignore this subtlety.

## B.3 HYPERPARAMETERS

Here, we first explain how the hyperparameter searches were performed. Then, in Appendix B.3.1 we provide the search space and selected values for every method. Finally, in Appendix B.3.2 we study the sensitivity of PEC and other methods to hyperparameters.

To select hyperparameters for our benchmarking experiments presented in Tables 1 and 2, for every combination of method, dataset, and task split, we perform a separate grid search. For every considered set of hyperparameters, we perform one experiment with a single random seed. We then select the hyperparameter values that yield the highest final validation set accuracy, and we use those values for our benchmarking experiments with ten different random seeds.

While it has been argued that in continual learning, extensive hyperparameter searches are not desired and instead hyperparameters should be decided on the fly (Chaudhry et al., 2019a), we perform the searches to ensure that the baseline methods get as good performance as possible, even if in practice they could not be tuned so well. We note that PEC does not introduce any method-specific hyperparameters. Moreover, for all PEC experiments in Section 4 we use batch size $= 1$, do not use data augmentations, and use linear learning decay. Only the learning rate varies between experiments. However, below we will show that even if the learning rate is not tuned and the default value of $0.001$ is used for all datasets, the performance of PEC is still strong. This indicates that our method can work well off the shelf.

### B.3.1 SEARCH SPACES AND SELECTED VALUES

For all methods except SLDA and Nearest mean, we tune the following hyperparameters:

- learning rate (lr) $\in \{0.00003, 0.0001, 0.0003, 0.001, 0.003, 0.01, 0.03\}$,
- batch size (bs) $\in \{1, 10, 32\}$,
- whether to use data augmentation (aug) $\in \{false, true\}$,
- whether to decay LR from the initial value to 0 linearly (decay) $\in \{false, true\}$.

Additional tuned hyperparameters for all methods, together with search spaces, are listed in Table 8.

We provide selected hyperparameter values for all methods in Table 10 for MNIST, in Table 11 for Balanced SVHN, in Table 12 for CIFAR-10, in Table 13 for CIFAR-100, and in Table 14 for miniImageNet.

### B.3.2 SENSITIVITY TO HYPERPARAMETER VALUES

First, we consider a setting where the learning rate cannot be tuned and we run PEC with the default learning rate value of $0.001$ for all datasets. Note that this means that only the architecture is varied between experiments (in order to match the proper parameter counts), and hence in such a case PEC can be treated as an off-the-shelf algorithm. We observe in Table 9 that PEC retains the strong performance in this case, and in particular it still outperforms other rehearsal-free methods, whose hyperparameters were tuned with a grid search, by wide margins (compare to Tables 1 and 2).

Table 8: Method-specific hyperparameters together with associated search spaces.

| Method | Additional hyperparameters |
|---|---|
| ER | — |
| A-GEM | — |
| DER | distillation loss coef. $\alpha \in \{0.1, 0.2, 0.5, 1\}$ |
| DER++ | distillation loss coef. $\alpha \in \{0.1, 0.2, 0.5, 1\}$
regular replay loss coef. $\beta \in \{0.1, 0.2, 0.5, 1\}$ |
| X-DER | distillation loss coef. $\alpha \in \{0.3, 0.6\}$
regular replay loss coef. $\beta \in \{0.8, 0.9\}$
consistency loss coef. $\lambda \in \{0.05, 0.1\}$
past/future logits constraint coef. $\eta \in \{0.001, 0.01\}$
constant for past/future logits constraint $m \in \{0.3, 0.7\}$ |
| ER-ACE | — |
| iCaRL | — |
| BiC | — |
| EWC | regularization coef. $\lambda \in \{0.1, 1, 10, 100, 1000\}$ |
| SI | regularization coef. $c \in \{0.5, 1, 2, 5\}$
damping coefficient $\xi \in \{0.9, 1\}$ |
| LwF | regularization coef. $\alpha \in \{0.5, 1, 2, 5, 10\}$ |
| Labels trick | — |
| Nearest mean | — |
| SLDA | identity matrix mixing coef. $\epsilon \in \{0.0001, 0.0003, 0.001, 0.003, 0.01, 0.03, 0.1, 0.3, 0.5, 0.8\}$ |
| VAE-GC | — |
| PEC | — |

Table 9: Performance comparison of PEC with tuned learning rates versus PEC using default learning rate of 0.001. Reported is the final average accuracy (in %), with mean and standard error from 10 seeds.

| Method | MNIST | Balanced SVHN | CIFAR-10 | CIFAR-100 | miniImageNet |
|---|---|---|---|---|---|
| PEC | 92.31 ($\pm$ 0.13) | 68.70 ($\pm$ 0.16) | 58.94 ($\pm$ 0.09) | 26.54 ($\pm$ 0.11) | 14.90 ($\pm$ 0.08) |
| PEC, $lr = 0.001$ | 90.51 ($\pm$ 0.16) | 64.25 ($\pm$ 0.18) | 57.38 ($\pm$ 0.12) | 26.54 ($\pm$ 0.11) | 14.90 ($\pm$ 0.08) |

Additionally, we look at how the performance of ER, DER++, VAE-GC, and PEC varies when we manipulate learning rates; the rest of the hyperparameters remain as selected originally. The results in Figure 4 indicate that the dependence of PEC on learning rate tuning is comparable to that of other methods.

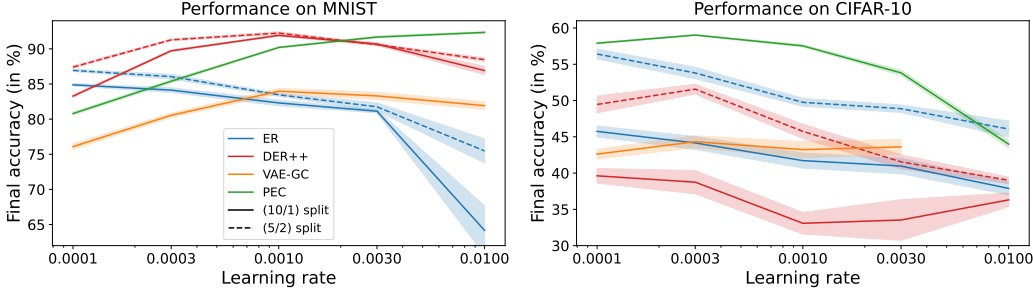

Figure 4: CIL performance for varying learning rate. Plotted are the means (solid or dashed lines) and standard errors (shaded areas) from 10 seeds. For PEC and VAE-GC, results are the same for both task splits, and hence single curves are shown. In case of VAE-GC experiment on CIFAR-10 with learning rate 0.01, runs failed because of numerical instabilities.

Table 10: Selected hyperparameters for MNIST dataset.

| Method | MNIST (10/1) | MNIST (5/2) |
|---|---|---|
| ER | $lr = 0.0003$, bs $= 1$, decay $= true$ | $lr = 0.0001$, bs $= 1$, decay $= true$ |
| A-GEM | $lr = 0.0003$, bs $= 32$, decay $= true$ | $lr = 0.001$, bs $= 10$, decay $= false$ |
| DER | $lr = 0.001$, bs $= 10$, decay $= true$, $\alpha = 1$ | $lr = 0.001$, bs $= 10$, decay $= true$, $\alpha = 1$ |
| DER++ | $lr = 0.001$, bs $= 10$, decay $= true$, $\alpha = 1$, $\beta = 1$ | $lr = 0.001$, bs $= 10$, decay $= true$, $\alpha = 0.5$, $\beta = 1$ |
| X-DER | $lr = 0.001$, bs $= 10$, decay $= false$, $\alpha = 0.3$, $\beta = 0.8$, $\lambda = 0.05$, $\eta = 0.01$, $m = 0.3$ | $lr = 0.003$, $bs = 10$, decay $= false$, $\alpha = 0.3$, $\beta = 0.9$, $\lambda = 0.05$, $\eta = 0.01$, $m = 0.7$ |
| ER-ACE | $lr = 0.001$, bs $= 10$, decay $= false$ | $lr = 0.001$, bs $= 10$, decay $= false$ |
| iCaRL | $lr = 0.003$, bs $= 10$, decay $= true$ | $lr = 0.003$, bs $= 10$, decay $= false$ |
| BiC | $lr = 0.001$, bs $= 10$, decay $= false$ | $lr = 0.001$, bs $= 10$, decay $= false$ |
| Labels trick | $lr = 0.001$, bs $= 1$, decay $= false$ | $lr = 0.0001$, bs $= 32$, decay $= false$ |
| EWC + LT | $lr = 0.0003$, bs $= 32$, decay $= true$, $\lambda = 1000$ | $lr = 0.0003$, bs $= 32$, decay $= false$, $\lambda = 1000$ |
| SI + LT | $lr = 0.003$, bs $= 10$, decay $= true$, $c = 1$, $\xi = 0.9$ | $lr = 0.003$, bs $= 1$, decay $= true$, $c = 0.5$, $\xi = 1$ |
| LwF + LT | $lr = 0.003$, bs $= 32$, decay $= true$, $\alpha = 5$ | $lr = 0.001$, bs $= 10$, decay $= true$, $\alpha = 5$ |
| Nearest mean | — | |
| SLDA | $\epsilon = 0.1$ | |
| VAE-GC | $lr = 0.003$, bs $= 32$, decay $= false$ | |
| PEC | $lr = 0.01$, bs $= 1$, decay $= true$ | |

Table 11: Selected hyperparameters for Balanced SVHN dataset.

| Method | Balanced SVHN (10/1) | Balanced SVHN (5/2) |
|---|---|---|
| ER | $lr = 0.003$, bs $= 1$, aug $= true$, decay $= true$ | $lr = 0.0001$, bs $= 1$, aug $= true$, decay $= true$ |
| A-GEM | $lr = 0.001$, bs $= 32$, aug $= true$, decay $= true$ | $lr = 0.01$, bs $= 10$, aug $= false$, decay $= false$ |
| DER | $lr = 0.0003$, bs $= 10$, aug $= true$, decay $= true$, $\alpha = 0.5$ | $lr = 0.0003$, bs $= 10$, aug $= false$, decay $= false$, $\alpha = 1$ |
| DER++ | $lr = 0.0003$, bs $= 10$, aug $= true$, decay $= true$, $\alpha = 0.2$, $\beta = 1$ | $lr = 0.0003$, bs $= 10$, aug $= true$, decay $= true$, $\alpha = 0.5$, $\beta = 1$ |
| X-DER | $lr = 0.0001$, bs $= 10$, aug $= false$, decay $= false$, $\alpha = 0.3$, $\beta = 0.8$, $\lambda = 0.1$, $\eta = 0.01$, $m = 0.7$ | $lr = 0.0001$, bs $= 10$, aug $= false$, decay $= true$, $\alpha = 0.6$, $\beta = 0.8$, $\lambda = 0.05$, $\eta = 0.01$, $m = 0.3$ |
| ER-ACE | $lr = 0.0001$, bs $= 10$, aug $= true$, decay $= true$ | $lr = 0.0003$, bs $= 10$, aug $= false$, decay $= true$ |
| iCaRL | $lr = 0.0001$, bs $= 10$, aug $= false$, decay $= false$ | $lr = 0.0001$, bs $= 10$, aug $= false$, decay $= false$ |
| BiC | $lr = 0.0001$, bs $= 10$, aug $= false$, decay $= true$ | $lr = 0.0003$, bs $= 10$, aug $= false$, decay $= true$ |
| Labels trick | $lr = 0.003$, bs $= 32$, aug $= false$, decay $= true$ | $lr = 0.00003$, bs $= 32$, aug $= false$, decay $= true$ |
| EWC + LT | $lr = 0.001$, bs $= 32$, aug $= false$, decay $= false$, $\lambda = 1000$ | $lr = 0.0001$, bs $= 32$, aug $= false$, decay $= false$, $\lambda = 0.1$ |
| SI + LT | $lr = 0.01$, bs $= 32$, aug $= false$, decay $= false$, $c = 1$, $\xi = 1$ | $lr = 3e - 05$, bs $= 10$, aug $= false$, decay $= true$, $c = 2$, $\xi = 1$ |
| LwF + LT | $lr = 0.003$, bs $= 10$, aug $= false$, decay $= true$, $\alpha = 1$ | $lr = 0.0003$, bs $= 10$, aug $= false$, decay $= true$, $\alpha = 10$ |
| Nearest mean | — | |
| SLDA | $\epsilon = 0.1$ | |
| VAE-GC | $lr = 0.0003$, bs $= 32$, aug $= false$, decay $= false$ | |
| PEC | $lr = 0.0001$, bs $= 1$, aug $= false$, decay $= true$ | |

Table 12: Selected hyperparameters for CIFAR-10 dataset.

| Method | CIFAR-10 (10/1) | CIFAR-10 (5/2) |
|---|---|---|
| ER | lr = 0.003, bs = 1, aug = $true$, decay = $true$ | lr = 0.0001, bs = 1, aug = $true$, decay = $false$ |
| A-GEM | lr = 0.003, bs = 32, aug = $false$, decay = $false$ | lr = 0.0003, bs = 10, aug = $false$, decay = $false$ |
| DER | lr = 0.0001, bs = 10, aug = $false$, decay = $true$, $\alpha = 0.5$ | lr = 0.0001, bs = 10, aug = $true$, decay = $false$, $\alpha = 0.5$ |
| DER++ | lr = 0.0001, bs = 10, aug = $false$, decay = $false$, $\alpha = 0.1$, $\beta = 0.5$ | lr = 0.0003, bs = 10, aug = $false$, decay = $false$, $\alpha = 1$, $\beta = 1$ |
| X-DER | lr = 0.0001, bs = 10, aug = $true$, decay = $false$, $\alpha = 0.3$, $\beta = 0.8$, $\lambda = 0.1$, $\eta = 0.01$, $m = 0.3$ | lr = 0.0001, bs = 10, aug = $false$, decay = $false$, $\alpha = 0.6$, $\beta = 0.8$, $\lambda = 0.05$, $\eta = 0.001$, $m = 0.3$ |
| ER-ACE | lr = 0.001, bs = 10, aug = $false$, decay = $false$ | lr = 0.0003, bs = 10, aug = $true$, decay = $true$ |
| iCaRL | lr = 0.0001, bs = 10, aug = $false$, decay = $true$ | lr = 0.0003, bs = 10, aug = $false$, decay = $true$ |
| BiC | lr = 0.0001, bs = 10, aug = $false$, decay = $true$ | lr = 0.0001, bs = 10, aug = $false$, decay = $false$ |
| Labels trick | lr = 0.03, bs = 1, aug = $false$, decay = $false$ | lr = 0.00003, bs = 10, aug = $false$, decay = $true$ |
| EWC + LT | lr = 0.0003, bs = 32, aug = $false$, decay = $true$, $\lambda = 1000$ | lr = 0.0001, bs = 10, aug = $false$, decay = $true$, $\lambda = 1000$ |
| SI + LT | lr = 0.0003, bs = 32, aug = $false$, decay = $false$, $c = 1$, $\xi = 1$ | lr = $3e - 05$, bs = 32, aug = $false$, decay = $false$, $c = 2$, $\xi = 1$ |
| LwF + LT | lr = 0.003, bs = 32, aug = $false$, decay = $false$, $\alpha = 10$ | lr = 0.0003, bs = 10, aug = $false$, decay = $true$, $\alpha = 10$ |
| Nearest mean | — | |
| SLDA | $\epsilon = 0.3$ | |
| VAE-GC | lr = 0.001, bs = 32, aug = $false$, decay = $true$ | |
| PEC | lr = 0.0003, bs = 1, aug = $false$, decay = $true$ | |

Table 13: Selected hyperparameters for CIFAR-100 dataset.

| Method | CIFAR-100 (100/1) | CIFAR-100 (10/10) |
|---|---|---|
| ER | lr = 0.003, bs = 1, aug = $true$, decay = $false$ | lr = 0.003, bs = 1, aug = $true$, decay = $false$ |
| A-GEM | lr = 0.00003, bs = 32, aug = $false$, decay = $false$ | lr = 0.01, bs = 10, aug = $false$, decay = $true$ |
| DER | lr = 0.00003, bs = 10, aug = $true$, decay = $false$, $\alpha = 0.1$ | lr = 0.01, bs = 10, aug = $false$, decay = $false$, $\alpha = 1$ |
| DER++ | lr = 0.01, bs = 10, aug = $true$, decay = $false$, $\alpha = 0.1$, $\beta = 1$ | lr = 0.01, bs = 10, aug = $true$, decay = $true$, $\alpha = 1$, $\beta = 1$ |
| X-DER | lr = 0.01, bs = 10, aug = $true$, decay = $false$, $\alpha = 0.3$, $\beta = 0.9$, $\lambda = 0.1$, $\eta = 0.01$, $m = 0.7$ | lr = 0.01, bs = 10, aug = $false$, decay = $false$, $\alpha = 0.6$, $\beta = 0.9$, $\lambda = 0.05$, $\eta = 0.01$, $m = 0.3$ |
| ER-ACE | lr = $3e-05$, bs = 32, aug = $true$, decay = $true$ | lr = 0.0003, bs = 10, aug = $false$, decay = $false$ |
| iCaRL | lr = 0.0001, bs = 10, aug = $true$, decay = $true$ | lr = 0.03, bs = 10, aug = $false$, decay = $false$ |
| BiC | lr = 0.01, bs = 10, aug = $true$, decay = $false$ | lr = 0.03, bs = 10, aug = $false$, decay = $true$ |
| Labels trick | lr = 0.0001, bs = 1, aug = $true$, decay = $false$ | lr = 0.03, bs = 32, aug = $false$, decay = $true$ |
| EWC + LT | lr = 0.001, bs = 32, aug = $false$, decay = $false$, $\lambda = 1$ | lr = 0.01, bs = 10, aug = $false$, decay = $false$, $\lambda = 10$ |
| SI + LT | lr = 0.001, bs = 10, aug = $false$, decay = $false$, $c = 2$, $\xi = 1$ | lr = $3e-05$, bs = 32, aug = $false$, decay = $false$, $c = 5$, $\xi = 0.9$ |
| LwF + LT | lr = $3e-05$, bs = 32, aug = $false$, decay = $true$, $\alpha = 0.5$ | lr = 0.0003, bs = 32, aug = $false$, decay = $true$, $\alpha = 10$ |
| Nearest mean | — | |
| SLDA | $\epsilon = 0.5$ | |
| VAE-GC | lr = 0.003, bs = 32, aug = $false$, decay = $true$ | |
| PEC | lr = 0.001, bs = 1, aug = $false$, decay = $true$ | |

Table 14: Selected hyperparameters for miniImageNet dataset.

| Method | miniImageNet (100/1) | miniImageNet (20/5) |
|---|---|---|
| ER | lr = 0.003, bs = 1, aug = $true$, decay = $false$ | lr = 0.01, bs = 1, aug = $true$, decay = $false$ |
| A-GEM | lr = 0.0003, bs = 32, aug = $true$, decay = $true$ | lr = 0.00003, bs = 32, aug = $false$, decay = $false$ |
| DER | lr = 0.003, bs = 10, aug = $true$, decay = $true$, $\alpha = 1$ | lr = 0.0001, bs = 10, aug = $true$, decay = $false$, $\alpha = 1$ |
| DER++ | lr = 0.003, bs = 10, aug = $false$, decay = $true$, $\alpha = 0.1$, $\beta = 1$ | lr = 0.001, bs = 10, aug = $true$, decay = $false$, $\alpha = 1$, $\beta = 1$ |
| X-DER | lr = 0.001, bs = 10, aug = $true$, decay = $false$, $\alpha = 0.6$, $\beta = 0.9$, $\lambda = 0.1$, $\eta = 0.01$, $m = 0.7$ | lr = 0.001, bs = 10, aug = $false$, decay = $false$, $\alpha = 0.3$, $\beta = 0.8$, $\lambda = 0.1$, $\eta = 0.001$, $m = 0.3$ |
| ER-ACE | lr = $3e - 05$, bs = 32, aug = $false$, decay = $true$ | lr = 0.0001, bs = 10, aug = $false$, decay = $false$ |
| iCaRL | lr = 0.00003, bs = 10, aug = $false$, decay = $false$ | lr = 0.0001, bs = 10, aug = $true$, decay = $false$ |
| BiC | lr = 0.00003, bs = 10, aug = $false$, decay = $false$ | lr = 0.0001, bs = 10, aug = $false$, decay = $false$ |
| Labels trick | lr = 0.003, bs = 32, aug = $true$, decay = $false$ | lr = 0.00003, bs = 10, aug = $true$, decay = $false$ |
| EWC + LT | lr = 0.003, bs = 32, aug = $false$, decay = $true$, $\lambda = 10$ | lr = 0.003, bs = 10, aug = $false$, decay = $false$, $\lambda = 1$ |
| SI + LT | lr = 0.03, bs = 1, aug = $false$, decay = $true$, $c = 0.5$, $\xi = 0.9$ | lr = $3e - 05$, bs = 10, aug = $false$, decay = $true$, $c = 2$, $\xi = 0.9$ |
| LwF + LT | lr = 0.0003, bs = 32, aug = $false$, decay = $true$, $\alpha = 5$ | lr = $3e - 05$, bs = 32, aug = $false$, decay = $false$, $\alpha = 10$ |
| Nearest mean | — | |
| SLDA | $\epsilon = 0.8$ | |
| VAE-GC | lr = 0.003, bs = 32, aug = $false$, decay = $true$ | |
| PEC | lr = 0.001, bs = 1, aug = $false$, decay = $true$ | |

### B.4 DETAILS ON BALANCING STRATEGIES

In Section 4.4, we discussed the issues of comparability of PEC class scores and the impact of dataset imbalance. We introduced strategies to alleviate the effect of data imbalance, and here we provide additional details about these.

In Oracle balancing and Buffer balancing, we use CMA-ES (Hansen & Ostermeier, 2001) to find scalars for re-scaling class scores. This evolutionary optimization algorithm maintains a Gaussian distribution, which is iteratively updated to fit values that yield high fitness scores. We tuned hyperparameters for this algorithm in our initial experiments and then used fixed values everywhere, which we provide here: initial standard deviation $= 0.001$, population size $= 100$, weight decay $= 0$. For each class $c$, we parameterize the scalar that we search for as $s_c = \exp(l_c)$ and we search for real values $l_c$, since we want to find positive values for the scalars.

In Equal budgets, we use the same number of gradient descent updates for each class. If we are allowed multiple passes over the dataset, then we can set some fixed value $N_U$ and use this many updates per class. In the single-pass-through-data setting, we cut the number of updates for each class to the minimal number of examples for any class.

### B.5 DATA AUGMENTATIONS

Here, we describe the data augmentations used in our experiments. Note that for every method we decide whether to use these data augmentations or not based on grid search; for PEC we never use these.

- MNIST: no augmentations are used.
- Balanced SVHN: random crop to original size with padding $= 4$.
- CIFAR-10, CIFAR-100, and miniImageNet: random crop to original size with padding $= 4$, random horizontal flip.

## C ADDITIONAL EXPERIMENTS

### C.1 ADDITIONAL COMPARISONS IN ANOTHER EXPERIMENTAL SETUP

Here, to further test the robustness of PEC, we perform additional empirical comparisons using an experimental setting based on van de Ven et al. (2021). First, we describe the experimental setup in Appendix C.1.1 and additional baselines in Appendix C.1.2, and then we proceed to the results in Appendix C.1.3.

### C.1.1 EXPERIMENTAL SETUP FOR APPENDIX C.1 ONLY

We largely follow the experimental setup from van de Ven et al. (2021). In the implementation of PEC, we match the parameter counts and the number of optimization steps with those used for VAE-GC in van de Ven et al. (2021).

We compare PEC against a comprehensive list of CIL methods; we provide explanations for the methods that were not included in our main text experiments in Appendix C.1.2.

For some datasets, namely MNIST (Deng, 2012), Fashion-MNIST (Xiao et al., 2017), and CIFAR-10 (Krizhevsky et al., 2009), we directly use raw image data. For others, namely CIFAR-100 (Krizhevsky et al., 2009) and CORe50 (Lomonaco & Maltoni, 2017), we use pretrained frozen feature extractors, which are applied on the raw data before further processing; we follow van de Ven et al. (2021), using the same feature extractors.

Below we briefly summarize the experimental setup for the particular datasets:

- **MNIST** and **Fashion-MNIST**: we use $(5/2)$ task split. Optimization for 2000 steps per task, or 1000 steps per class, with batch size $= 128$. MLP is used as an architecture, no feature extractor is used. Parameter count per class for VAE-GC and PEC methods: $150K$.
- **CIFAR-10**: we use $(5/2)$ task split. Optimization for 5000 steps per task, or 2500 steps per class, with batch size $= 256$. Simple networks with convolutions and linear layers are used as architectures, no feature extractor is used. Parameter count per class for VAE-GC and PEC methods: $350K$.

- **CIFAR-100**: we use $(10/10)$ task split. Optimization for 5000 steps per task, or 500 steps per class, with batch size $= 256$. MLP is used as an architecture, a simple convolutional network pre-trained on CIFAR-10 is used as a feature extractor. Parameter count per class for VAE-GC and PEC methods: $180K$.
- **CORe50**: we use $(5/2)$ task split. For this dataset, we only perform one pass through the data during training. We use batch size $= 1$. MLP is used as an architecture, ResNet18 (He et al., 2016) pre-trained on ImageNet (Deng et al., 2009) is used as a feature extractor. Parameter count per class for VAE-GC and PEC methods: $170K$.

In all experiments, we use the Adam (Kingma & Ba, 2014) optimizer. We set the learning rate to the standard value of $0.001$, with the following exceptions:

- for the CORe50 experiments, LR is set to $0.0003$ for PEC and $0.0001$ for other methods;
- for the CIFAR-100 experiments, LR is set to $0.0001$ for discriminative approaches and standard $0.001$ otherwise.

Additionally, for all PEC experiments, the learning rate decays linearly to $0$ through the training.

For VAE-GC we use $10K$ importance samples to estimate likelihoods.

The method-specific hyperparameters of SI, EWC and AR1 are selected based on the grid search described in van de Ven et al. (2021). While for the experiments in Tables 1 and 2 the shrinkage parameter $\epsilon$ of SLDA was tuned, for the experiments in this section the default value of $0.0001$ is used.

For VAE-GC experiments on MNIST, we additionally include batch normalization in the architecture, since we observed it improves performance. Additionally, in order to ensure a fair comparison with VAE-GC, the best-performing baseline in these experiments, we run a grid search for this method where we incorporate the following elements that are used with PEC: linear learning rate decay and GELU activation function (Hendrycks & Gimpel, 2016). These did not improve the results for VAE-GC.

### C.1.2 ADDITIONAL METHODS

We provide short descriptions and references for baselines used in this section that were not present in our main text experiments:

*Bias-correction* methods attempt to mitigate imbalances between classes coming from different tasks.

- CopyWeights with Re-init (**CWR**) (Lomonaco & Maltoni, 2017) maintains the frozen representation network after the initial task, and for each consecutive task, only updates the part of the final linear layer corresponding to classes from that task.
- **CWR+** (Maltoni & Lomonaco, 2019) improves over CWR by introducing different initialization and normalization schemes.
- **AR1** (Maltoni & Lomonaco, 2019) builds on CWR+ but enables representations to shift, regularizing them additionally with SI.

*Generative replay* uses generative models to produce samples for pseudo-rehearsal.

- Deep Generative Replay (**DGR**) (Shin et al., 2017) leverages VAE to produce examples from past tasks and uses them as additional data for training.

### C.1.3 RESULTS

The results are presented in Table 15. Note that the experimental setup used here involves multiple epochs and, in most cases, higher parameter counts than in the main text experiments. As a consequence, the advantage of PEC over VAE-GC is less pronounced. Nonetheless, PEC retains its strong performance and outperforms or matches all the baselines considered here on all datasets. This corroborates its robustness to experimental conditions.

### C.2 PERFORMANCE OF REHEARSAL-FREE METHODS WITHOUT LABELS TRICK

In the main text, we reported the results of rehearsal-free methods with Labels trick added on top. For completeness, here we provide results without the Labels trick. Results for the one-class-per-task and multiple-classes-per-task are presented in Table 16 and Table 17, respectively.

Table 15: Empirical evaluation of PEC against a suite of CIL baselines, in experimental setups based on van de Ven et al. (2021), see Appendix C.1. Final average accuracy (in %) is reported, with mean and standard error from 10 seeds. Methods are separated into the following categories: baselines, regularization-based, bias-correction, generative replay, generative classification, PEC. **FE**: pre-trained feature extractor is used.

| Method | MNIST (5/2) | Fashion-MNIST (5/2) | CIFAR-10 (5/2) | CIFAR-100 (10/10), **FE** | CORe50 (5/2), **FE** |
|---|---|---|---|---|---|
| *Fine-tuning* | *19.92 (± 0.02)* | *19.73 (± 0.25)* | *18.72 (± 0.27)* | *7.98 (± 0.10)* | *18.52 (± 0.29)* |
| *Joint* | *98.27 (± 0.03)* | *89.50 (± 0.08)* | *82.61 (± 0.12)* | *51.42 (± 0.13)* | *71.50 (± 0.20)* |
| EWC | 20.00 (± 0.07) | 19.75 (± 0.24) | 18.59 (± 0.29) | 8.34 (± 0.08) | 18.59 (± 0.34) |
| SI | 19.98 (± 0.11) | 19.57 (± 0.42) | 18.15 (± 0.31) | 8.54 (± 0.08) | 18.65 (± 0.27) |
| CWR | 30.18 (± 2.59) | 21.80 (± 2.25) | 18.26 (± 1.38) | 22.16 (± 0.69) | 40.17 (± 0.82) |
| CWR+ | 38.61 (± 2.93) | 19.80 (± 2.77) | 19.06 (± 1.79) | 9.48 (± 0.32) | 40.35 (± 0.98) |
| AR1 | 44.27 (± 2.50) | 29.14 (± 2.17) | 24.60 (± 1.08) | 17.49 (± 0.34) | 46.77 (± 0.93) |
| Labels Trick | 33.05 (± 1.32) | 20.78 (± 1.90) | 20.14 (± 0.40) | 24.02 (± 0.39) | 42.81 (± 1.08) |
| DGR | 91.30 (± 0.53) | 69.18 (± 1.70) | 17.73 (± 1.58) | 15.14 (± 0.20) | 56.67 (± 1.21) |
| SLDA | 87.30 (± 0.02) | 81.50 (± 0.01) | 38.39 (± 0.04) | 44.48 (± 0.00) | **70.80 (± 0.00)** |
| VAE-GC | 96.29 (± 0.04) | 82.90 (± 0.06) | 55.92 (± 0.13) | 49.46 (± 0.08) | 70.74 (± 0.08) |
| **PEC** | **96.73 (± 0.04)** | **85.22 (± 0.07)** | **57.75 (± 0.09)** | **49.80 (± 0.09)** | **70.81 (± 0.11)** |

Table 16: Performance of rehearsal-free methods without the Labels trick in the one-class-per-task case. Final average accuracy (in %) is reported, with mean and standard error from 10 seeds. This table complements Table 1 in the main text.

| Method | MNIST (10/1) | Balanced SVHN (10/1) | CIFAR-10 (10/1) | CIFAR-100 (100/1) | miniImageNet (100/1) |
|---|---|---|---|---|---|
| EWC | 10.09 (± 0.00) | 9.92 (± 0.08) | 10.55 (± 0.44) | 1.00 (± 0.00) | 0.99 (± 0.02) |
| SI | 12.73 (± 0.99) | 10.00 (± 0.00) | 10.08 (± 0.06) | 1.10 (± 0.04) | 0.97 (± 0.07) |
| LwF | 11.83 (± 0.58) | 9.94 (± 0.03) | 10.12 (± 0.09) | 0.93 (± 0.02) | 0.97 (± 0.04) |

Table 17: Performance of rehearsal-free methods without the Labels trick in the multiple-classes-per-task case. Final average accuracy (in %) is reported, with mean and standard error from 10 seeds. This table complements Table 2 in the main text.

| Method | MNIST (5/2) | Balanced SVHN (5/2) | CIFAR-10 (5/2) | CIFAR-100 (10/10) | miniImageNet (20/5) |
|---|---|---|---|---|---|
| EWC | 20.10 (± 0.18) | 12.99 (± 1.43) | 12.39 (± 0.86) | 6.81 (± 0.12) | 1.88 (± 0.20) |
| SI | 21.46 (± 0.48) | 19.13 (± 0.02) | 18.75 (± 0.02) | 6.42 (± 0.12) | 2.94 (± 0.11) |
| LwF | 21.49 (± 1.86) | 23.88 (± 0.62) | 20.14 (± 0.18) | 7.75 (± 0.13) | 3.31 (± 0.03) |

Table 18: Performance with parameter budget of $1.1M$, for the one-class-per-task case. For discriminative methods, Narrow ResNet18 architecture is used. Final average accuracy (in %) is reported, with mean and standard error from 10 seeds.

| Method | Buffer | Balanced SVHN (10/1) | CIFAR-10 (10/1) | CIFAR-100 (100/1) | miniImageNet (100/1) |
|---|---|---|---|---|---|
| DER++ | ✓ | **77.07** ($\pm$ **1.01**) | 30.98 ($\pm$ 2.45) | 3.73 ($\pm$ 0.12) | 2.59 ($\pm$ 0.12) |
| X-DER | ✓ | 71.30 ($\pm$ 0.47) | 36.46 ($\pm$ 2.25) | 12.17 ($\pm$ 0.39) | **10.36** ($\pm$ **0.31**) |
| Labels trick (LT) | - | 9.87 ($\pm$ 0.06) | 10.05 ($\pm$ 0.11) | 1.03 ($\pm$ 0.03) | 1.03 ($\pm$ 0.05) |
| EWC + LT | - | 10.04 ($\pm$ 0.07) | 9.72 ($\pm$ 0.29) | 0.93 ($\pm$ 0.03) | 1.00 ($\pm$ 0.05) |
| SI + LT | - | 9.99 ($\pm$ 0.05) | 9.52 ($\pm$ 0.33) | 0.99 ($\pm$ 0.06) | 1.05 ($\pm$ 0.05) |
| LwF + LT | - | 9.92 ($\pm$ 0.13) | 9.83 ($\pm$ 0.28) | 0.98 ($\pm$ 0.01) | 1.00 ($\pm$ 0.00) |
| EWC | - | 9.91 ($\pm$ 0.07) | 9.31 ($\pm$ 0.44) | 0.98 ($\pm$ 0.08) | 1.01 ($\pm$ 0.06) |
| SI | - | 10.00 ($\pm$ 0.00) | 10.00 ($\pm$ 0.00) | 1.00 ($\pm$ 0.03) | 0.96 ($\pm$ 0.05) |
| LwF | - | 9.97 ($\pm$ 0.07) | 9.91 ($\pm$ 0.04) | 1.02 ($\pm$ 0.05) | 0.99 ($\pm$ 0.01) |
| **PEC (ours)** | - | 60.17 ($\pm$ 0.21) | **53.58** ($\pm$ **0.31**) | **19.67** ($\pm$ **0.36**) | **10.84** ($\pm$ **0.18**) |

Table 19: Performance with parameter budget of $1.1M$, for the many-classes-per-task case. For discriminative methods, Narrow ResNet18 architecture is used. Final average accuracy (in %) is reported, with mean and standard error from 10 seeds.

| Method | Buffer | Balanced SVHN (5/2) | CIFAR-10 (5/2) | CIFAR-100 (10/10) | miniImageNet (20/5) |
|---|---|---|---|---|---|
| DER++ | ✓ | **84.44** ($\pm$ **0.36**) | 42.86 ($\pm$ 0.70) | 16.83 ($\pm$ 0.39) | 9.86 ($\pm$ 0.29) |
| X-DER | ✓ | 71.30 ($\pm$ 3.70) | 52.16 ($\pm$ 1.08) | **22.91** ($\pm$ **0.38**) | **16.48** ($\pm$ **0.33**) |
| Labels trick (LT) | - | 26.56 ($\pm$ 1.23) | 22.87 ($\pm$ 0.67) | 8.03 ($\pm$ 0.17) | 3.99 ($\pm$ 0.06) |
| EWC + LT | - | 25.28 ($\pm$ 1.06) | 23.33 ($\pm$ 0.96) | 8.01 ($\pm$ 0.14) | 3.38 ($\pm$ 0.11) |
| SI + LT | - | 28.12 ($\pm$ 1.18) | 21.20 ($\pm$ 0.72) | 7.39 ($\pm$ 0.18) | 4.16 ($\pm$ 0.07) |
| LwF + LT | - | 41.37 ($\pm$ 1.48) | 38.08 ($\pm$ 0.79) | 10.92 ($\pm$ 0.29) | 3.29 ($\pm$ 0.18) |
| EWC | - | 17.51 ($\pm$ 0.87) | 15.53 ($\pm$ 0.36) | 2.94 ($\pm$ 0.09) | 2.18 ($\pm$ 0.05) |
| SI | - | 17.99 ($\pm$ 0.67) | 18.46 ($\pm$ 0.06) | 5.31 ($\pm$ 0.09) | 2.91 ($\pm$ 0.04) |
| LwF | - | 26.14 ($\pm$ 0.18) | 20.99 ($\pm$ 0.30) | 5.89 ($\pm$ 0.03) | 2.86 ($\pm$ 0.04) |
| **PEC (ours)** | - | 60.17 ($\pm$ 0.21) | **53.58** ($\pm$ **0.31**) | 19.67 ($\pm$ 0.36) | 10.84 ($\pm$ 0.18) |

## C.3 Performance evaluation with lower parameter budgets

In this section, we perform additional experiments, where we replace the variant of ResNet18 that we used in our main text experiments with a narrower version as is used by Lopez-Paz & Ranzato (2017); Chaudhry et al. (2019b); Aljundi et al. (2019a); Caccia et al. (2021); Soutif-Cormerais et al. (2023). We call this variant Narrow ResNet18. We note that the only difference between the architecture we used in the main experiments and the Narrow ResNet18 is the width, with the former having a width factor of $64$ and the latter of $20$. We also adjust the parameter count for PEC to match the $1.1M$ count of the Narrow ResNet18. For this experiment, we performed limited hyperparameter searches: for each method, we take the hyperparameters that were used for the variant with the bigger (default) parameter budget and tune only learning rate. Results are reported in Table 18 for the one-class-per-task case and in Table 19 for the many-classes-per-task case. We can see that PEC retains its strong performance in this regime and still outperforms all non-rehearsal baselines in all considered cases. PEC's performance looks favorably even compared to challenging rehearsal-based approaches, DER++ and X-DER, as PEC outperforms them in $4$ out of $8$ considered cases.

## C.4 Performance when varying number of ensemble members

Here we study how the performance changes if we allow for an ensemble with independent student-teacher pairs. For simplicity, we did not include this possibility in our description of PEC and experiments in the main text. However, it is a natural variant to consider, especially given the theoretical support from Section 3.2. Results in Figure 5 demonstrate increased performance when using more than one student-teacher pair. The gains are roughly comparable to those when the width of single networks increases, see Figure 3.

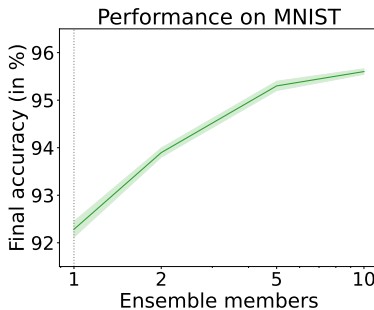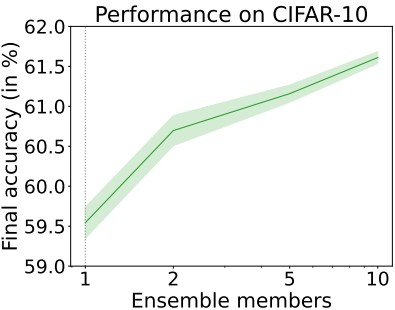

Figure 5: CIL performance for varying number of ensemble members. Plotted are the means (solid lines) and standard errors (shaded areas) from 10 seeds.

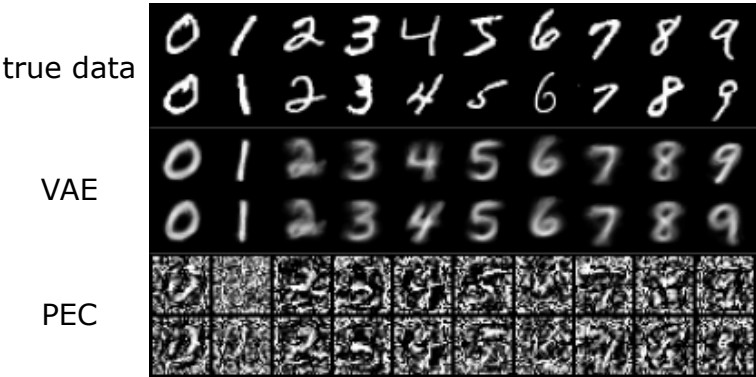

Figure 6: Original data from the MNIST dataset is compared to generations from VAE and PEC modules used in our experiments. Each column shows samples for a different class.

### C.5 IS PEC SECRETLY A GENERATIVE MODEL?

Here we examine if one can generate samples from datasets' class distributions given PEC modules. Since the score of a PEC module is low for examples from its corresponding class, we try to generate samples by starting from a random input and then minimizing the PEC score for a given class by gradient descent on inputs. Specifically, we utilize Adam with the learning rate of $0.00003$, and perform $30K$ updates. Generated samples are presented in Figure 6 and compared against samples from the class-specific VAE modules of VAE-GC, as well as original data. Interestingly, samples from PEC bear a resemblance to respective digits. However, the quality of generations is very poor.

### C.6 RESULTS FOR METHODS WITH MEMORY BUFFER FOR VARYING BUFFER SIZE

In our main text experiments, when comparing against methods that leverage a memory buffer, we focused on a moderate buffer size of $500$. Here we include comparisons for the case of small and big memory buffers, containing respectively 100 and 5000 examples, following regimes from Buzzega et al. (2020). For these experiments, we perform the same hyperparameter searches as for the main text results, see Appendix B.3.

Results for the case of one class per task are presented in Table 20 and for the case of multiple classes per task are presented in Table 21. As expected, the performance of these methods drops significantly as we move to the smaller buffer, increasing the advantage of PEC. In the case of big buffer size, PEC is still competitive in the one-class-per-task setting and scores the best for CIFAR-100 and miniImageNet datasets, while in the multiple-classes-per-task setting its advantage mostly fades away. This is not surprising, as replay-based methods are increasingly similar to the joint training as the buffer size grows.

Table 20: Results for the varying buffer size for the methods that use memory buffer in case of one class per task. Final average accuracy (in %) is reported, with mean and standard error from 10 seeds. Results for PEC are included to ease comparison.

| Method | Buffer size | MNIST (10/1) | Balanced SVHN (10/1) | CIFAR-10 (10/1) | CIFAR-100 (100/1) | miniImageNet (100/1) |
|---|---|---|---|---|---|---|
| ER | 100 | 68.67 (± 1.04) | 46.53 (± 1.98) | 28.39 (± 0.80) | 5.74 (± 0.17) | 2.76 (± 0.15) |
| A-GEM | 100 | 63.90 (± 1.21) | 10.35 (± 0.18) | 11.40 (± 0.57) | 0.90 (± 0.04) | 1.50 (± 0.07) |
| DER | 100 | 81.27 (± 0.64) | 57.67 (± 2.42) | 25.33 (± 0.76) | 1.00 (± 0.03) | 1.01 (± 0.04) |
| DER++ | 100 | 81.41 (± 0.39) | 59.11 (± 1.89) | 23.33 (± 1.23) | 2.55 (± 0.33) | 2.45 (± 0.19) |
| X-DER | 100 | 80.18 (± 0.61) | 48.64 (± 3.86) | 30.93 (± 0.44) | 7.45 (± 0.23) | 4.92 (± 0.17) |
| iCaRL | 100 | 75.45 (± 0.54) | 22.16 (± 2.99) | 28.99 (± 0.39) | 5.06 (± 0.16) | 3.46 (± 0.16) |
| BiC | 100 | 64.48 (± 1.59) | 14.39 (± 1.13) | 12.48 (± 0.72) | 3.12 (± 0.28) | 1.16 (± 0.05) |
| ER | 500 | 84.42 (± 0.26) | 74.80 (± 0.69) | 40.59 (± 1.17) | 12.47 (± 0.32) | 5.72 (± 0.24) |
| A-GEM | 500 | 59.84 (± 0.82) | 9.93 (± 0.03) | 10.19 (± 0.11) | 1.01 (± 0.03) | 1.11 (± 0.07) |
| DER | 500 | 91.65 (± 0.11) | 76.45 (± 0.71) | 40.03 (± 1.52) | 1.01 (± 0.05) | 0.99 (± 0.01) |
| DER++ | 500 | 91.88 (± 0.19) | 80.66 (± 0.48) | 35.61 (± 2.43) | 6.17 (± 0.38) | 1.40 (± 0.09) |
| X-DER | 500 | 82.97 (± 0.14) | 70.65 (± 0.61) | 43.16 (± 0.46) | 15.55 (± 0.14) | 8.22 (± 0.35) |
| iCaRL | 500 | 83.08 (± 0.28) | 57.18 (± 0.94) | 37.80 (± 0.42) | 5.74 (± 0.14) | 7.52 (± 0.14) |
| BiC | 500 | 86.00 (± 0.37) | 70.69 (± 0.52) | 35.86 (± 0.38) | 6.44 (± 0.28) | 1.46 (± 0.05) |
| ER | 5000 | 92.34 (± 0.41) | 82.06 (± 0.68) | 43.90 (± 0.59) | 20.76 (± 0.65) | 6.33 (± 0.31) |
| A-GEM | 5000 | 59.27 (± 0.96) | 9.98 (± 0.04) | 11.32 (± 0.59) | 0.97 (± 0.06) | 1.03 (± 0.04) |
| DER | 5000 | 93.74 (± 0.08) | 85.47 (± 0.63) | 51.41 (± 0.63) | 0.95 (± 0.03) | 0.97 (± 0.04) |
| DER++ | 5000 | 95.01 (± 0.07) | 88.36 (± 0.25) | 49.51 (± 2.04) | 9.22 (± 0.78) | 1.56 (± 0.13) |
| X-DER | 5000 | 86.23 (± 0.09) | 78.54 (± 0.24) | 52.32 (± 1.23) | 22.33 (± 0.80) | 3.31 (± 0.15) |
| iCaRL | 5000 | 88.02 (± 0.21) | 81.97 (± 0.28) | 61.13 (± 0.22) | 8.64 (± 0.21) | 14.17 (± 0.52) |
| BiC | 5000 | 93.11 (± 0.17) | 86.89 (± 0.25) | 62.84 (± 0.40) | 2.84 (± 0.06) | 2.95 (± 0.14) |
| **PEC (ours)** | - | 92.31 (± 0.13) | 68.70 (± 0.16) | 58.94 (± 0.09) | 26.54 (± 0.11) | 14.90 (± 0.08) |

Table 21: Results for the varying buffer size for the methods that use memory buffer in case of multiple classes per task. Final average accuracy (in %) is reported, with mean and standard error from 10 seeds. Results for PEC are included to ease comparison.

| Method | Buffer size | MNIST (5/2) | Balanced SVHN (5/2) | CIFAR-10 (5/2) | CIFAR-100 (10/10) | miniImageNet (20/5) |
|---|---|---|---|---|---|---|
| ER | 100 | 60.89 (± 1.42) | 58.07 (± 0.45) | 38.75 (± 0.63) | 10.78 (± 0.34) | 4.37 (± 0.14) |
| A-GEM | 100 | 57.45 (± 1.26) | 20.40 (± 0.85) | 18.10 (± 0.15) | 6.07 (± 0.05) | 2.67 (± 0.16) |
| DER | 100 | 81.68 (± 0.37) | 70.40 (± 1.25) | 37.09 (± 0.83) | 8.62 (± 0.18) | 3.95 (± 0.08) |
| DER++ | 100 | 81.52 (± 0.71) | 66.64 (± 1.67) | 40.31 (± 0.73) | 10.74 (± 0.19) | 6.41 (± 0.23) |
| X-DER | 100 | 84.19 (± 0.94) | 74.79 (± 0.81) | 51.60 (± 0.81) | 19.44 (± 0.24) | 8.83 (± 0.81) |
| iCaRL | 100 | 74.53 (± 0.52) | 58.45 (± 0.74) | 48.05 (± 0.34) | 10.87 (± 0.14) | 9.78 (± 0.13) |
| BiC | 100 | 68.10 (± 1.44) | 27.84 (± 0.88) | 23.34 (± 1.07) | 8.42 (± 0.23) | 4.08 (± 0.02) |
| ER | 500 | 86.55 (± 0.17) | 80.00 (± 0.54) | 57.49 (± 0.54) | 18.15 (± 0.17) | 6.67 (± 0.21) |
| A-GEM | 500 | 50.82 (± 1.36) | 19.71 (± 0.49) | 18.21 (± 0.14) | 6.43 (± 0.06) | 2.47 (± 0.18) |
| DER | 500 | 91.64 (± 0.11) | 78.22 (± 0.94) | 43.98 (± 1.20) | 7.69 (± 0.27) | 3.86 (± 0.04) |
| DER++ | 500 | 92.30 (± 0.14) | 86.50 (± 0.43) | 50.61 (± 0.80) | 19.88 (± 0.25) | 10.50 (± 0.42) |
| X-DER | 500 | 81.91 (± 0.53) | 73.88 (± 1.59) | 54.15 (± 1.92) | 27.84 (± 0.28) | 14.65 (± 0.40) |
| iCaRL | 500 | 80.41 (± 0.45) | 73.65 (± 0.57) | 54.33 (± 0.33) | 13.31 (± 0.16) | 13.46 (± 0.20) |
| BiC | 500 | 88.08 (± 0.81) | 77.65 (± 0.45) | 46.74 (± 0.83) | 14.12 (± 0.26) | 6.90 (± 0.19) |
| ER | 5000 | 94.60 (± 0.11) | 87.62 (± 0.52) | 63.55 (± 0.28) | 26.71 (± 0.36) | 9.14 (± 0.37) |
| A-GEM | 5000 | 46.52 (± 1.37) | 18.33 (± 0.52) | 18.49 (± 0.13) | 6.66 (± 0.05) | 2.92 (± 0.12) |
| DER | 5000 | 95.02 (± 0.06) | 84.08 (± 0.84) | 55.57 (± 0.72) | 7.12 (± 0.21) | 3.85 (± 0.06) |
| DER++ | 5000 | 95.70 (± 0.07) | 91.29 (± 0.15) | 61.41 (± 0.57) | 24.07 (± 0.40) | 5.31 (± 0.41) |
| X-DER | 5000 | 87.78 (± 0.47) | 79.79 (± 1.22) | 59.80 (± 0.52) | 34.58 (± 0.30) | 16.94 (± 0.58) |
| iCaRL | 5000 | 87.56 (± 0.29) | 85.03 (± 0.17) | 63.85 (± 0.23) | 14.49 (± 0.11) | 19.39 (± 0.18) |
| BiC | 5000 | 94.55 (± 0.14) | 88.38 (± 0.19) | 69.84 (± 0.22) | 25.42 (± 0.21) | 19.94 (± 0.16) |
| **PEC (ours)** | - | 92.31 (± 0.13) | 68.70 (± 0.16) | 58.94 (± 0.09) | 26.54 (± 0.11) | 14.90 (± 0.08) |

Table 22: Performance comparison of PEC for various initialization schemes. Reported is the final average accuracy (in %), with mean and standard error from 10 seeds.

| Method | MNIST | CIFAR-10 |
|---|---|---|
| PEC, Kaiming (vanilla) | 92.31 ($\pm$ 0.13) | 58.94 ($\pm$ 0.09) |
| PEC, Xavier | 91.70 ($\pm$ 0.13) | 58.02 ($\pm$ 0.14) |
| PEC, random uniform from $[-0.001, 0.001]$ | 88.52 ($\pm$ 0.22) | 57.33 ($\pm$ 0.22) |
| PEC, random uniform from $[-0.01, 0.01]$ | 91.73 ($\pm$ 0.09) | 57.60 ($\pm$ 0.20) |
| PEC, random uniform from $[-0.1, 0.1]$ | 92.01 ($\pm$ 0.14) | 57.46 ($\pm$ 0.20) |

## C.7 PEC WITH DIFFERENT INITIALIZATION SCHEMES

We verify how much the performance of PEC depends on the chosen initialization scheme. We study Kaiming (He et al., 2015) (our default choice), Xavier (Glorot & Bengio, 2010), and random uniform with different ranges. For each choice of initialization scheme, we use it to initialize the parameters of both the teacher and the student networks. The results in Table 22 indicate that the performance of PEC is relatively stable with respect to the way the networks are initialized.

## C.8 PEC WITH LINEAR ARCHITECTURES

Here we further explore architectural choices for PEC. We compare PEC with our vanilla architecture, which has a single hidden layer with a nonlinear activation function, to PEC with linear networks that contain no activation function. We consider two variants. For both variants, the described changes are applied to both the teacher and the student networks. In *Removed nonlinearity*, we take the vanilla PEC architecture and remove the nonlinear activation function, resulting in a linear network that retains some inductive biases of the original architecture (e.g. the use of convolutions). In *Single linear*, we just take a single linear layer with the output size chosen to match the parameter count of the original architecture. Results in Table 23 indicate that typically vanilla PEC performs substantially better than the linear variants. Interestingly, for Balanced SVHN, Single linear performs better than vanilla. A reason for this might be the suboptimality of the used convolutional architecture, which we did not tune extensively.

## C.9 PEC VERSUS ENSEMBLE DISCRIMINATIVE CLASSIFIER

Here we report additional experiments in which we test whether the strong performance of PEC could be explained by network ensembling (i.e., training a separate network per task), rather than that it is due to the PEC classification rule. To test this, we compare PEC with various variants of an 'ensemble discriminative classifier baseline'. For this ensemble baseline, we train for each task a separate network equipped with a task-specific softmax layer, and during inference we take the argmax of the predicted scores across the softmax layers of the different models. We consider three variants of this ensemble baseline. In *Ensemble, full*, we use for each task a separate copy of the original discriminative network (i.e., two-layer MLP for MNIST, ResNet18 for the other datasets). Note that this baseline uses $T$ times more parameters than PEC, with $T$ the number of tasks. In *Ensemble, small*, we use for each task a version of the original discriminative network (i.e., two-layer MLP for MNIST, ResNet18 for the other datasets), but with the width of the networks reduced in order to match the parameter count of PEC and other baselines. Finally, in *Ensemble, PEC-architecture*, we use for each task the same network architecture as for PEC, except with a softmax output layer instead of the $d$-dimensional linear output layer and with the width of the hidden layer adjusted in order to match the parameter count of PEC. This last variant is the most direct comparison, because it is as close to PEC as possible, except that it does not use the PEC classification rule.

To give these ensemble baselines the best possible chance, we run these three variants in the easier multiple-classes-per-task setting. (In preliminary experiments we confirmed that these ensemble baselines indeed perform substantially worse in the single-class-per-task case.) The results are reported in Table 24. We see that, on all five datasets, PEC substantially outperforms all three variants of the ensemble baseline. This provides strong evidence that the PEC classification rule is critical for the strong performance of PEC.

Table 23: Performance comparison of vanilla PEC versus variants of PEC with linear networks. Reported is the final average accuracy (in %), with mean and standard error from 10 seeds.

| Method | MNIST | Balanced SVHN | CIFAR-10 | CIFAR-100 | miniImageNet |
|---|---|---|---|---|---|
| PEC | 92.31 ($\pm$ 0.13) | 68.70 ($\pm$ 0.16) | 58.94 ($\pm$ 0.09) | 26.54 ($\pm$ 0.11) | 14.90 ($\pm$ 0.08) |
| PEC, Removed nonlinearity | 92.41 ($\pm$ 0.12) | 61.85 ($\pm$ 0.20) | 46.19 ($\pm$ 0.14) | 19.41 ($\pm$ 0.11) | 10.55 ($\pm$ 0.11) |
| PEC, Single linear | 80.39 ($\pm$ 0.24) | 71.08 ($\pm$ 0.18) | 40.04 ($\pm$ 0.14) | 13.21 ($\pm$ 0.10) | 7.30 ($\pm$ 0.11) |

Table 24: Comparison of PEC with several versions of the ensemble discriminative classifier baseline, for the multiple-classes-per-task case. Reported is the final average accuracy (in %), with mean and standard error from 10 seeds.

| Method | MNIST | Balanced SVHN | CIFAR-10 | CIFAR-100 | miniImageNet |
|---|---|---|---|---|---|
| PEC | 92.31 ($\pm$ 0.13) | 68.70 ($\pm$ 0.16) | 58.94 ($\pm$ 0.09) | 26.54 ($\pm$ 0.11) | 14.90 ($\pm$ 0.08) |
| Ensemble, full | 55.47 ($\pm$ 1.38) | 52.17 ($\pm$ 1.70) | 41.15 ($\pm$ 1.00) | 21.48 ($\pm$ 0.21) | 8.83 ($\pm$ 0.19) |
| Ensemble, small | 48.04 ($\pm$ 2.16) | 47.78 ($\pm$ 0.98) | 38.31 ($\pm$ 0.77) | 17.58 ($\pm$ 0.22) | 7.75 ($\pm$ 0.23) |
| Ensemble, PEC-architecture | 51.09 ($\pm$ 1.71) | 47.70 ($\pm$ 0.28) | 36.39 ($\pm$ 0.13) | 24.52 ($\pm$ 0.17) | 12.82 ($\pm$ 0.18) |

# D    DETAILED THEORETICAL SUPPORT FOR PEC

## D.1    PROOFS OF THE PROPOSITIONS FROM MAIN TEXT

**Proposition 1** (Proposition 1 from Ciosek et al. (2020), rephrased). *Let $H$ be a Gaussian Process and $X$ a dataset. Then the following inequality holds:*

$$s_\infty^{H,X}(x^*) \geq \sigma(H_X)(x^*). \tag{7}$$

*Proof.* We can directly use Proposition 1 from (Ciosek et al., 2020, Section 4.1, Appendix F.1) with $f$ from the original proposition statement being equal to $H$, and function $h$ from the original proposition statement being equal to $\varphi(X, \hat{h}(X), x^*) := g^{\hat{h}, X}(x^*)$, for any $\hat{h} \sim H$. Note that $\tilde{\sigma}_\mu^2(x_\star)$ and $\sigma_X^2(x_\star)$ from the original statement correspond to our $s_\infty^{H,X}(x^*)$ and $\sigma(H_X)(x^*)$, respectively. $\square$

**Lemma 1** (Lemma 4 from Ciosek et al. (2020), rephrased). *Let $h : \mathbb{R}^K \to \mathbb{R}^M$ be a function with the domain restricted to $\{x : ||x||_\infty \leq 1\}$, $U \geq 0$ be a constant such that $||h(x)||_\infty \leq U$ for all $x$ in the domain, and $\mathcal{X}$ be a data distribution with support in $\{x : ||x||_\infty \leq 1\}$. Furthermore, assume that $K \geq 3$, and that our procedure for training neural networks $g^{h,X}$ returns Lipschitz functions with constant $L$. Then, for a dataset $X := \{x_1, \ldots, x_N\} \sim \mathcal{X}$, the following inequality holds with probability $1 - \delta$:*

$$\mathbb{E}_{x^* \sim \mathcal{X}} \left[ \frac{1}{M} ||g^{h,X}(x^*) - h(x^*)||^2 \right] \leq \frac{1}{MN} \sum_{i=1}^{N} ||g^{h,X}(x_i) - h(x_i)||^2 \\ + LU \cdot O\left( \frac{1}{\sqrt[K]{N}} \sqrt{\frac{\log(1/\delta)}{N}} \right). \tag{8}$$

*Proof.* Let us directly use Lemma 4 from (Ciosek et al., 2020, Appendix F.2), with $f$ from the original formulation being $h$, and $h_{X_f}$ from the original formulation being $g^{h,X}$. $\square$

**Proposition 2.** *Let $H$ be a Gaussian Process, $\mathcal{X}$ be a data distribution, and $X := \{x_1, \ldots, x_N\} \sim \mathcal{X}$ a finite dataset. Assume that our class of approximators is such that for any $h \sim H$, $h$ can be approximated by $g^{h,X}$ so that, for arbitrarily large training sets, the training error is smaller than $\epsilon > 0$ with probability at least $1 - \gamma$, $\gamma > 0$. Then, under technical assumptions, for any $\delta > 0$, it holds with probability at least $1 - \delta - \gamma$ that*

$$\mathbb{E}_{x^* \sim \mathcal{X}} \left[ s_1^{H,X}(x^*) \right] \leq \epsilon + \kappa_N, \tag{9}$$

*where $\kappa_N$ goes to 0 as $N \to \infty$.*

*Proof.* The technical assumptions that are needed are those of Lemma 1. Let $h_1 \sim H$ be the sample used by $s_1^{H,X}$ and use Lemma 1 for $h_1$. On the right side of Equation 8 we have the training error plus another factor that goes to 0 when $N$ grows, which we denote by $\kappa_N$. Thus, combining Equation 8 (inequality holding with probability $1 - \delta$) with the assumption on the training error (inequality holding with probability $1 - \gamma$), we get that both positive events occur with probability at least $1 - \delta - \gamma$, in which case we obtain

$$\mathbb{E}_{x^* \sim \mathcal{X}} \left[ s_1^{H,X}(x^*) \right] = \mathbb{E}_{x^* \sim \mathcal{X}} \left[ ||g^{h,X}(x^*) - h(x^*)||^2 \right] \leq$$

$$\frac{1}{N} \sum_{i=1}^{N} ||g^{h,X}(x_i) - h(x_i)||^2 + LU \cdot O \left( \frac{1}{\sqrt[K]{N}} \sqrt{\frac{\log(1/\delta)}{N}} \right) \leq \epsilon + \kappa_N,$$

which we wanted to prove. $\qquad\square$

### D.2 ADDITIONAL PROPOSITION ABOUT GP POSTERIOR VARIANCE RULE

In Section 3.2, we provided an intuitive argument for the use of the GP posterior variance rule. Here we present an additional proposition that shows convergence to 0 of scores for in-distribution data.

We assume the setting without the observation noise. In such a case, the formula for a posterior variance of a Gaussian Process $H := \mathcal{GP}(\mu, k)$ at a point $x^*$ given dataset $(X, y)$ is as follows (Rasmussen & Williams, 2006):

$$\sigma(H_X)(x^*) = k(x^*, x^*) - k(x^*, X)^T k(X, X)^{-1} k(X, x^*). \tag{10}$$

**Proposition 3.** *Let $H$ be a Gaussian Process with continuous kernel function $k$, $X$ be a dataset sampled according to $\mathcal{X}$, and $x^* \sim \mathcal{X}$. Then the posterior variance $\sigma(H_X)(x^*)$ goes to 0 in the limit of infinite data.*

*Proof.* Since $k$ is continuous, so is the function $\psi(x) := k(x^*, x^*) - k(x^*, x)k(x, x)^{-1}k(x, x^*)$. Therefore, for any $\epsilon > 0$, there exists $\delta > 0$ so that $||\psi(x_0) - \psi(x^*)|| < \epsilon$ whenever $||x_0 - x^*|| < \delta$. We also see that

$$||\psi(x_0) - \psi(x^*)|| = ||k(x^*, x^*) - k(x^*, x_0)k(x_0, x_0)^{-1}k(x_0, x^*) - 0||,$$

which, using equation 10, is a posterior variance with respect to the dataset $\{x_0\}$. In the limit of infinite data, $X$ will contain a point $x_0$ such that $||x_0 - x^*|| < \delta$. In such a case, we will have

$$\sigma(H_X)(x^*) = \sigma((H_{\{x_0\}})_{X \setminus \{x_0\}})(x^*) \leq \sigma(H_{\{x_0\}})(x^*) < \epsilon.$$

Here, first, we decompose computation of the posterior $H_X$ into taking posteriors with respect to $\{x_0\}$ and $X \setminus \{x_0\}$, and then we use the fact that the prior variance is not bigger than the posterior variance. The resulting inequality concludes the proof. $\qquad\square$

## E RESOURCES

For running experiments, we used Slurm-based clusters with heterogenous hardware, including nodes with A100 GPUs, V100 GPUs, and CPUs only. We note that because of the single-pass-through-data setting and relatively small dataset sizes, our experiments were feasible to run in all of the mentioned hardware setups. To produce our main benchmarking results in Tables 1 and 2, 1300 runs are needed, each taking on average 21.5 minutes on an A100 GPU, which makes 466 hours in total. During working on the project, we executed over $65K$ experimental runs.

