# OpenReview forum: "Prediction Error-based Classification for Class-Incremental Learning"
_ICLR.cc/2024/Conference — ICLR 2024 poster_

### Official Review · Reviewer_HWad · 2023-10-18

**Soundness:** 2 fair
**Presentation:** 2 fair
**Contribution:** 3 good
**Rating:** 5
**Confidence:** 4

**Summary:**

This paper introduces a method called Prediction Error-based classification (PEC) for Online Class-Incremental Learning. Unlike existing methods in the continual learning literature, relying upon discriminative and generative classification paradigms, it presents a novel classification approach for continual learning. The method involves fitting a shallow predictor network for each task class to match a single random prior network. At inference, the final classification is determined by selecting the class with the smallest error between all the fitted models up to the current task and the output representation of the random prior network. The authors emphasize that the classification rule they employ is theoretically justified as an approximation of a rule based on Gaussian Process posterior variance. This approach operates in an exemplar-free scenario, doesn't rely on a pre-trained network, and can handle the  problem of having a single class per task. The authors conducted empirical experiments, showing  that their approach enhances performance compared to exemplar-free baselines and exemplar-based methods relying on generative or discriminative classification paradigms. The authors highlight that their approach exhibits class-imbalancing issues, and they propose two methods to mitigate this problem: Buffer Balancing, which relies on an evolutionary optimization algorithm and an exemplar memory buffer, and Equal Budget, which equalizes the number of training iterations for different classes.

**Strengths:**

*  The introduction is well-written and they clearly motivate why they propose prediction error-based classification.
*  The novel usage of a Random Prior Network and multiple predictor network for online class incremental classification.
*  The PEC's ability to work in exemplar-free, from scratch, one class-per-task settings is a valuable feature. These settings, especially the combination of these ones, are infrequently assessed by the literature.

**Weaknesses:**

The major weakness of this paper lies in the experimental evaluation where the authors assess the performance of the proposed method:


* **SOTA Architecture Comparison**:  The  authors have designed the PEC predictor models to align  with the parameter count of reference literature networks.  Specifically for the evaluation of SOTA methods, the PEC authors employ simpler networks architecture for  SVHN, MNIST, while they employ the standard Resnet18  for Cifar10, Split-Cifar100 and the miniImagenet dataset. These latter tasks are considered more challenging in comparison to SVHN and MNIST.  My primary concern revolves around the use of the standard Resnet18 that the authors employ for evaluating SOTA methods. Training the standard Resnet18 can be difficult, especially when the number of training iterations is limited (i.e. in online setting), primarly due to the large number of parameters (11.2M as reported in the  table 6 of the Appendix B.1 of the paper). In response to this challenge, recent sota online approaches rely upon a **reduced** version of Resnet18, which consists approximately on 1.1M of parameters [g] [h] [i] [l] [m] [n]. This reduced network is commonly favored for online continual learning due to its computational efficiency and ease of optimization. Utilizing the reduced Resnet18 as a reference is essential to ensure a fair comparison with the existing literature, especially since PEC uses shallow models easier to train than Resnet18. It's worth noting that doing so would require the overall number of parameters in the PEC models to be reduced to 1.1 million to match the reference architecture.


* **Discriminative Exemplar-Free Class Incremental (EFCIL) Comparison**: Comparing PEC, an online exemplar-free method, with other common exemplar-free approaches is crucial. In the main paper, only EWC [a] and Label Trick[b] are evaluated, with EWC designed for offline continual learning and not suitable for online settings. Other common weight regularization methods, such as MAS [c] and SI[d], are typically evaluated in the online continual learning literature, since they allows to compute online the weight importance. Moreover, LwF[e], regularizing output with knowledge distillation, is known to outperform weight regularization methods and is suitable for online settings.The absence of this comparison is noteworthy. The only other baseline provided by the authors in the main paper is "Label Trick"[b], involves training only the head of the current task. Nowadays, this is a common practice in exemplar-free approaches, since it has been shown to work better. All the exemplar-free implemented methods (included LwF) on FACIL (Masana et al. [f]) use this practice. All the exemplar-free method reported in the main paper must apply this trick. Moreover, in Appendix C.1.1, various EFCIL methods are presented, and it might be beneficial to move some of them to the main paper, implementing the Label Trick where applicable for a more comprehensive evaluation.



* **Discriminative Exemplar-Based Class Incremental (EBCIL) Comparison**: PEC is an **Online** method. When comparing it with exemplar-based method, it is essential to align the compared methods with the current **Online** EBCIL literature. ER, A-GEM, DER++, ICARL, already reported in the main paper,  are common comparison in the online literature. On the other hand, BiC is an **offline**  method. This is because it requires additional epochs on a validation set to calibrate the classifier(thus violating online incremental constraints). X-DER, a recent work, is original designed in the multi-epoch setting (i.e. offline) but has been adapted in this paper to work in online setting. To ensure a comprehensive comparison with the current online literature, it's important to consider additional relevant approaches.  A straightforward  baseline to provide is "ER + LwF". Notably, ER-ACE [g] has recently delivered state-of-the-art results in online continual learning, so it is advisable to include this comparison in the main paper. Other approaches that should be evaluated include SCR [l] and RAR [m]. For a more extensive list of relevant comparison, you can refer to  Cormerais et al. [n].

* **Memory Requirements & Inference FLOPS Computation**: Regarding the **Memory Requirements**, PEC requires a linear increase of parameters with the number of classes encountered. While the authors acknowledge this intrinsic challenge in their approach and have discussed it in the limitations section, a more detailed analysis is necessary. Specifically, it's essential to examine how many parameters are utilized by each predictor networks per dataset and by the teacher network, to understand how much the memory increases across the tasks. As for the **FLOPS comparison**, the authors have not provided an analysis of the number of Floating Point Operations (FLOPS) required by their model. Since online methods are intended to work in environments with hardware resource limitations, it is advisable to disclose the number of FLOPS for both backward and forward passes. For training, FLOPS evaluation can be carried out, as demonstrated by [o]. As regards the inference step, in Section 3.1 the authors say that only two forward passes are needed to obtain the classification scores – one for the a merged student, and one for the teacher network. While this is true, considering the online nature of their method and its intended use in resource-constrained settings, it is crucial to provide and compare the overall number of FLOPS required by the forward pass for both teacher and student networks. Therefore, it is necessary to determine how much the FLOPS of PEC method increases compared to current state-of-the-art methods


* **Buffer Balancing and Equal Budget**: The authors assert that their approach is afflicted by class imbalancing.However, they do not provide any theoretical motivation for this claim; they hypothesize that the issue may be linked to the number of gradient updates, as mentioned in Section 4.4 of the main paper. In cases where no theoretical motivation is available, it is necessary to conduct a detailed analysis that elucidates the factors contributing to this behavior. To address the class imbalancing issue, they authors introduce two strategies in the experimental section: Buffer Balancing and Equal Budget Strategies. While the easier Equal Budget Strategy it is well-explained in the paper (and in supplementary Section B.2), the **Buffer balancing**, based on CMA-ES optimization is considerable more complex and difficult to grasp. Therefore, additional details on how this optimization algorithm is used for online-class incremental learning should be provided to enhance understanding. Finally, a natural question arises: How well do existing methods perform in the class imbalancing setting compared to PEC?

**References**

[a] James Kirkpatrick et al. "Overcoming catastrophic forgetting in neural networks," in Proceedings of the National Academy of Sciences, 2017.

[b] Chen Zeno  et al, "Task Agnostic Continual Learning Using Online Variational Bayes," 2019.

[c] T. Aljundi et al. , "Memory Aware Synapses: Learning What (not) to Forget," in  ECCV 2018.

[d] Zenke et al. (2017). Continual Learning Through Synaptic Intelligence. Proceedings of machine learning research.

[e] Z. Li, D. Hoiem. "Learning without Forgetting," in IEEE Transactions on Pattern Analysis and Machine Intelligence, 2018.

[f] Masana, M. et al. "Class-Incremental Learning: Survey and Performance Evaluation on Image Classification," in IEEE Transactions on Pattern Analysis and Machine Intelligence, 2023.


[g] Lucas Caccia  et al, "New Insights on Reducing Abrupt Representation Change in Online Continual Learning," in International Conference on Learning Representations, 2022

[h] Aljundi et al. , "Online Continual Learning with Maximal Interfered Retrieval," in Advances in Neural Information Processing Systems, 2019.

[i] Chaudhry, A., et al. "Continual Learning with Tiny Episodic Memories" 2019.

[l] Zheda Mai et al. "Supervised Contrastive Replay: Revisiting the Nearest Class Mean Classifier in Online Class-Incremental Continual Learning," in 2021 IEEE/CVF Conference on Computer Vision and Pattern Recognition Workshops (CVPRW),  2021

[m] Kumari, L., et al, "Retrospective Adversarial Replay for Continual Learning," in Advances in Neural Information Processing Systems, 2022


[n] Soutif-Cormerais  et al, "A Comprehensive Empirical Evaluation on Online Continual Learning," in Proceedings of the IEEE/CVF International Conference on Computer Vision (ICCV) Workshops, 2023,

[o] Ghunaim, Y., et al, "Real-Time Evaluation in Online Continual Learning: A New Hope," in Conference on Computer Vision and Pattern Recognition (CVPR), 2023.

**Questions:**

My major concerns and questions are in the weakness section. However, there are other experiments that can further support the evaluation of the proposed approach and can help understand the performance of the  proposed method:

* **Ablation on the proposed method - PEC vs Ensemble Discriminative Classifiers**: PEC takes inspiration from Random Network Distillation (Burda et al. 2018), originally  designed for detecting novel states in reinforcement learning, and from work of Ciosek et al. 2020. [q], which focused on  uncertainty estimation for out-of-distribution (OOD) detection. Notably, Ciosek et al. demonstrated that Random Priors outperformed Network Ensembling. PEC adapts the concept of Random Prior to the  incremental learning setting, using a Random Prior Network and training a single prediction network per class. This naturally leads to the question of how well their proposed method performs in comparison of  ensembling with multiple discriminative classifiers. Ensembling multiple discriminative classifiers involves training small networks  equipped with a Softmax Layer for each task, and then taking the argmax of the prediction across the models. This evaluation can be conducted with multiple classes per task, which is relatively easier than the single-class per task scenario but quite common in the online continual learning community. To perform these experiments requiring a network per task, an architecture with the structure similar to the one provided by Zenke et al. ([d] Appendix A) can be employed.


* **Ablation on the proposed method - PEC vs Discriminative Classifier**: Although the reduced Resnet18 is used for comparison (as mentioned in the first point of weakness section), it is essential to note that training a deep network is inherently more challenging than training a shallow network, as performed by PEC. Additionally, when using a reduced ResNet18 architecture, a state-of-the-art online method should address the issue of Batch Normalization, which is currently a challenge in online class incremental learning [p]. PEC may have an advantage in this context since it employs shallow models for which batch normalization is not necessary.  To gain a deeper understanding of how well PEC performs compared to the discriminative methods, it would be valuable to evaluate some of the sota  methods (exemplar-free and exemplar-based) on SplitCifar100 or minImagenet datasets without using ResNet architecture. An option is to use the network provided  by Zenke et al. [d](Appendix A), not using batch normalization.This can help eliminate any bias stemming from architecture-specific issues in  incremental learning providing a better understanding of how well Prediction Error Based Classification performs in comparison to Discriminative classification.




**Summary Of The Review**

The paper introduces an adaptation of the Random Prior Network for Online Continual Learning. The theoretical foundation of this adaptation is not entirely novel, as it relies on the theory presented by Chosek et al. [q]. However, the application of the theory to the PEC classification framework in the context of online continual learning is interesting.

Unfortunately, I have significant concerns  pertaining the experimental evaluation. The paper lacks a comprehensive and properly aligned comparison with the current literature on online continual learning, as mentioned in Weakness Section Points 1, 2, and 3.  Additionally, the paper does not provide  comprehensive assessments of Memory Impact and FLOPS Comparison, as noted in Weakness Section Point 4. Finally to enhance the quality of the experimental evaluation and to understand how much the proposed method works well, it would be beneficial to include a more thorough ablation study of the proposed method, as suggested in Question Section Points 1 and 2.

Taking into account the above considerations, I believe that the paper still needs substantial work to be carried out, before it can be considered for publication. I thus believe that it shall not be accepted at the current issue of ICLR.

**References**

[d] Zenke et al. (2017). Continual Learning Through Synaptic Intelligence. Proceedings of machine learning research.

[p] Q. Pham, C. Liu, H. Steven, "Continual Normalization: Rethinking Batch Normalization for Online Continual Learning," in International Conference on Learning Representations, 2022.

[q] Ciosek, Kamil et al. “Conservative Uncertainty Estimation By Fitting Prior Networks.” International Conference on Learning Representations (2020).

---

> ### Author Response · Authors · 2023-11-16
> **Response, part 1/3**
>
> We thank the Reviewer for their very thorough review with many useful suggestions. We are encouraged that, in spite of the low overall score, the Reviewer finds our method novel and clearly motivated, and appreciates that it works in challenging setups. Below we address the specific points raised by the Reviewer with a substantial number of additional experiments, including new baselines, combinations of rehearsal-free methods with the “Labels Trick”, and experiments using the smaller ResNet. We believe these make our experimental section even more convincing and address the main concerns raised by the Reviewer.
>
> While we appreciate the thoroughness of the Reviewer and their efforts to ensure best practice in every aspect of the paper, we would also like to ask for some pragmatism. In particular, we would like to stress that our paper is not a survey paper. We propose a new and original method for continual learning, and we aim to provide convincing evidence that this method has merits. Papers that propose a novel method are typically expected to provide a representative set of baselines, not all possible baselines. We looked at how many baselines (not counting fine-tuning, joint, or multi-task training) are included in the main text of papers introducing the methods that we compare to. Results are: ER: 3, A-GEM: 8, DER/DER++: 11, X-DER: 10, iCaRL: 2, BiC: 3, EWC: 1, SI: 2, LwF: 4, SLDA: 7, VAE-GC: 10. Our paper (revised version) includes 15 baselines, more than any of the above. While it is true that the earlier papers had fewer possible baselines to compare to, we still think that this indicates that our experimental section is exceptionally strong.
>
> We would like to respectfully ask the Reviewer not to consider whether there are some experiments that could be added (because there always are) but rather whether we provide solid enough evidence for the competitive performance of PEC.
>
> Now we will address the specific points raised by the Reviewer one by one:
>
> **(1) SOTA Architecture Comparison**
>
> We thank the Reviewer for making this important point. We are happy to say that we ran additional experiments with the architecture requested by the Reviewer. Before we report on the experiments, let us describe and clarify the variants of ResNet that we are discussing. The 1.1M model mentioned by the Reviewer (let’s call it Narrow ResNet18) has two differences with respect to the original ResNet18:
> 1) the initial downscaling (stride 2 in the initial convolution and max pooling) is removed,
> 2) the width factor (affecting the number of filters in the convolutional layers across the network) is changed from 64 to 20.
>
> We used in our experiments the variant of ResNet18 from [1] (let’s call it DER-ResNet18), which incorporates the change 1) of Narrow ResNet18 but keeps the width of the original ResNet18. We apologize for not properly describing this in the initial submission; we have now revised the text to include this information, see Section 4.1 (we mark all revised fragments with orange). Importantly, the only difference between Narrow ResNet18 and DER-ResNet18 is width. The depth (which typically makes the optimization harder) does not change between the models. This suggests that the performance with DER-ResNet18 is expected to be better than with Narrow ResNet18, even in the online case. We verify this empirically below.
>
> In our additional experiment, reported in Appendix C.3, we rerun selected baselines with the Narrow ResNet18 architecture, as well as PEC with the adjusted 1.1M parameter budget. Due to limited time, we were not yet able to run all the baselines; however, we already included all rehearsal-free baselines (as they are most directly comparable with PEC) as well as DER++ and X-DER, two of the rehearsal-based methods that performed best in our original experiments. We will include all the baselines for the camera-ready version. Results in Tables 18 & 19 confirm that PEC retains its competitive performance with the smaller parameter budget. It still outperforms other rehearsal-free methods in every case, with wide margins. PEC is also able to outperform challenging rehearsal-based methods in 4 out of 8 cases. Furthermore, we can see that typically, the performance of a given method deteriorates when Narrow ResNet18 is used instead of DER-ResNet18. This provides an important sanity check for our original choice of architecture. We believe that these results fully address the issue raised by the Reviewer.
>
> Finally, we note that in our original submission, in Figure 2 we already partly addressed the issue, as we reported the performance of PEC vs several baselines when changing the parameter counts. For CIFAR-10 (bottom right figure), the left end of the $x$-axis roughly corresponds to the Narrow ResNet18.
>
> ***Note: this is part 1/3 of the response.***

---

> > ### Author Response · Authors · 2023-11-16
> > **Response, part 2/3**
> >
> > **(2) Discriminative Exemplar-Free Class Incremental (EFCIL) Comparison**
> >
> > We are happy to have been able to include additional baselines requested by the Reviewer, namely SI and LwF. If the Reviewer believes it is important, we will add MAS as well in the camera-ready version; the reason for not including it now is that this method was not readily available in our software framework. However, given the results of other regularization-based methods, it seems unlikely that MAS could challenge the performance of PEC. Additionally, we follow the Reviewer’s advice to add Labels Trick on top of all non-rehearsal methods. We now report versions with Labels Trick in our main text Tables 1 & 2, and report vanilla performances in Appendix C.2. After adding the requested baselines, PEC still outperforms rehearsal-free approaches in all cases, with an especially large lead in the one-class-per-task case.
> >
> > **(3) Discriminative Exemplar-Based Class Incremental (EBCIL) Comparison**
> >
> > We note that PEC is an exemplar-free method and we demonstrate that it outperforms exemplar-free methods in multiple cases (all that we consider). It is not our claim that PEC always outperforms exemplar-based methods, and in fact, in the paper, we show some cases where it does not. However, we experimentally show that PEC is still competitive against rehearsal-based methods using small replay buffers, and include strong recent baselines such as DER++ or X-DER. We think this is a major additional merit of our paper. However, we don’t see why this would require us to include an even vaster array of exemplar-based baselines (as PEC is exemplar-free!).
> >
> > That being said, we are happy to say that we included one additional baseline requested by the Reviewer, namely ER-ACE (results are included in Tables 1 & 2). This baseline does not change the conclusions of the paper, as PEC outperforms it in 8 out of 10 cases (all but Balanced SVHN experiments).
> >
> > **(4) Memory Requirements & Inference FLOPS Computation**
> >
> > Regarding memory requirements, parameter counts for all methods are disclosed in Appendix B.2, paragraph “Parameter counts”.
> >
> > Regarding FLOPS, in the revised version we added a report on the number of multiply-accumulate operations (MACs) used for a single inference forward pass, for PEC vs discriminative approaches, see Appendix B.2. For convenience, we include the results here as well:
> >
> > | Method         | MNIST | SVHN | CIFAR-10 | CIFAR-100 | miniImageNet |
> > |----------------|-------|------|----------|-----------|--------------|
> > | PEC            | 4.5M  | 352M | 352M     | 300M      | 1.9G         |
> > | discriminative | 90K   | 556M | 556M     | 556M      | 3.9G         |
> >
> > We see that except for the MNIST case, PEC uses fewer operations than the discriminative baseline. Intuitively, this is because PEC does a substantial part of its computation in a fully connected layer which has a much smaller compute-to-parameters density compared to convolutional layers which mostly constitute the ResNet architecture.
> > Unfortunately, we could not report numbers on backpropagation as we were not able to quickly find appropriate software for this. However, we note that in this case, the advantage of PEC should be even higher, as the backpropagation is not performed for the teacher, which constitutes a large part of our overall computation graph.
> >
> > **(5) Buffer Balancing and Equal Budget**
> >
> > As for the first part of the issue, we show for both SVHN and CIFAR-10 that (1) performance for the unbalanced version is bad, (2) performance for the balanced version is good, (3) if we remove the “imbalance” (e.g. by using our Equal budgets strategy), then the method recovers the good performance. Together, this constitutes evidence that class imbalance can cause the vanilla version of PEC to perform badly. We believe the fact that we clearly describe limitations like this one contributes to the scientific standard of our work; we think that a deeper analysis, although interesting, would be out of scope.
> >
> > As for explaining balancing strategies, we tried our best to do it in the original version of the paper; in the revised version, we include slight changes in Section 4.4. Regarding Buffer balancing, if the predictions need to be performed during the online training, the balancing procedure could be done either before each prediction or from time to time, to trade off the quality of computed balancing scalars and the computational complexity. We are happy to answer further questions if there are any.
> >
> > ***Note: this is part 2/3 of the response.***

---

> > > ### Author Response · Authors · 2023-11-16
> > > **Response, part 3/3**
> > >
> > > **(6) Ablation on the proposed method - PEC vs Ensemble Discriminative Classifiers**
> > >
> > > We think this is a nice idea for an additional experiment. Although interesting, we do not think this experiment is necessary to support our claims. Due to limited time during the discussion period, we won’t be able to implement it. Additionally, we argue that this approach would not work in the case of a single class per task, as there is nothing to discriminate between (similarly as in non-rehearsal baselines that we consider).
> > >
> > > **(7) Ablation on the proposed method - PEC vs Discriminative Classifier**
> > >
> > > Although this suggestion is interesting, we think it is out of the scope of the paper. We need to ensure that we give the best possible chance to the baselines to work as well as possible. We think we did this by performing vast hyperparameter searches and performing extensive experiments for many setups and datasets. Ablating choices like the one proposed by the Reviewer would move our publication closer to a survey paper, and it was not our intention to write one.
> > >
> > > ***We want to thank the Reviewer once again for their extremely detailed review containing many great suggestions. We think we were able to use those suggestions to improve our work and address all the major issues raised by the Reviewer. Given this, we ask the Reviewer to consider raising the score or, otherwise, point out any remaining issues.***
> > >
> > > *References*
> > >
> > > [1] P Buzzega, M Boschini, A Porrello, D Abati, S Calderara. Dark experience for general continual learning: a strong, simple baseline. NeurIPS 2020.
> > >
> > > ***Note: this is part 3/3 of the response.***

---

> ### Author Response · Authors · 2023-11-20
> **Follow-up**
>
> We wanted to follow up on our initial response, and ask if the Reviewer is satisfied by how we addressed the raised issues.
>
> Please let us know if there are any further questions.

---

> > ### Comment · Reviewer_HWad · 2023-11-21
> >
> > I appreciate the authors for their responses to most of my questions. I acknowledge their efforts in defending their work by providing additional comparisons, and I also appreciate the fact that the revised paper effectively highlights the changes, making the review process more straightforward. Adding the results for Narrow Resnet-18, in particular, allows for more comparisons within the online continual learning community.
> >
> > While I understand the authors' perspective that the paper should not become a survey paper, I still maintain that the suggestions I provided were necessary for a comprehensive understanding of the performance of the proposed method's components. The questions I posed in the "Question Section" were not intended to provide a chance for improvement to state-of-the-art approaches but rather to understand if the improvement of PEC was due to network ensembling or due to the **PEC Classification Rule**.
> >
> > By conducting these ablation studies, I still believe that the claims about discriminative classification and PEC in the paper would be stronger. Having fully read the other reviews for the present submission, my assessment is that in the current rebuttal, the authors have provided additional experiments, bringing the paper closer to the acceptance threshold but not beyond. I have adjusted my score accordingly and will discuss with the other reviewers and the AC in the next phase.

---

> > > ### Author Response · Authors · 2023-11-23
> > >
> > > We thank the Reviewer for engaging with our rebuttal. We are glad that the Reviewer appreciates our additional comparisons, and that they decided to raise their score.
> > >
> > > We also appreciate the additional comments about the questions that the Reviewer posed in the “Question Section”. We now understand better what the Reviewer intended with these. We are happy to say that we ran several additional experiments to address the question whether the strong performance of PEC is due to “network ensembling” or due to the “PEC classification rule”. In particular, we ran various versions of the “ensemble discriminative classifier baseline” that the Reviewer suggested. That is, for each task we train a separate network equipped with a task-specific softmax layer, and during inference we take the argmax of the predicted scores across the models. For this ensemble baseline, we consider three versions with different “base networks”:
> > > - **Ensemble-full**: we use for each task a separate copy of the original discriminative network (i.e., 2-layer MLP for MNIST, ResNet18 for other datasets). Note that this baseline uses $T$ times more parameters than PEC, with $T$ the number of tasks.
> > > - **Ensemble-small**: we use for each task a version of the original discriminative network (i.e., 2-layer MLP for MNIST, ResNet18 for other datasets), but with the width of the networks reduced in order to match the parameter count of PEC and other baselines.
> > > - **Ensemble-PECarchitecture**: we use for each task the same network architecture as for PEC, except with a softmax output layer instead of the $d$-dimensional linear layer and with the width of the hidden layer adjusted in order to match the parameter count of PEC. This last variant is the most direct comparison, because it is as close to PEC as possible, except that it does not use the PEC classification rule.
> > >
> > > As mentioned in our original rebuttal, this ensemble baseline is unlikely to work well in our challenging single-class-per-task setting. (We empirically verified with some initial experiments that this ensemble baseline indeed performs very badly in the single-class-per-task case.) Therefore, to give these ensemble baselines the best possible chance, we ran them in our easier multiple-classes-per-task setting. The results of these experiments are reported in Appendix C.9 (all new additions for this second rebuttal are indicated in blue). For convenience, we include the results here as well:
> > >
> > > | Method                     | MNIST          | Balanced SVHN  | CIFAR-10       | CIFAR-100      | miniImageNet   |
> > > |----------------------------|----------------|----------------|----------------|----------------|----------------|
> > > | PEC                        | 92.31 (± 0.13) | 68.70 (± 0.16) | 58.94 (± 0.09) | 26.54 (± 0.11) | 14.90 (± 0.08) |
> > > | Ensemble, full             | 55.47 (± 1.38) | 52.17 (± 1.70) | 41.15 (± 1.00) | 21.48 (± 0.21) | 8.83 (± 0.19)  |
> > > | Ensemble, small            | 48.04 (± 2.16) | 47.78 (± 0.98) | 38.31 (± 0.77) | 17.58 (± 0.22) | 7.75 (± 0.23)  |
> > > | Ensemble, PEC-architecture | 51.09 (± 1.71) | 47.70 (± 0.28) | 36.39 (± 0.13) | 24.52 (± 0.17) | 12.82 (± 0.18) |
> > >
> > > We see that, on all five datasets, PEC substantially outperforms all three variants of the ensemble baseline. ***We believe this provides strong and convincing evidence that the PEC classification rule is critical for the strong performance of PEC.***

---

### Official Review · Reviewer_pyfp · 2023-10-27

**Soundness:** 2 fair
**Presentation:** 2 fair
**Contribution:** 2 fair
**Rating:** 5
**Confidence:** 3

**Summary:**

The authors proposed the Prediction Error-based Classification (PEC) method in class-incremental learning, which mitigates the possible issue of data storage in the discriminative classification, and the issue of learning good generative models with limited data. To resolve these issues, the authors proposed to train simple class-wise generative (student) networks, which are used to replicate the results from the frozen random (teacher) network. The decision then is made by choosing the class with the smallest error between a student and the teacher network.

**Strengths:**

* The proposed PEC is a novel method as an efficient alternative to the generative models in class-incremental learning.
* The extensive experiments support the good performance of the proposed method in single-pass-through-data class-incremental learning.

**Weaknesses:**

* Since the student networks in the proposed method are decoupled from each other, the method ignores the intrinsic interdependence among classes.

* The "efficiency" claimed in the contribution is less empirically supported.  Furthermore, due to the individual design for student networks, is this a good plus when the number of classes increases?

* The notations are bad. In algorithms, the number of classes is $N$ but later it is denoted by $C$ in equation (1). In proposition (2), I believe $N$ denotes the sample size but it was denoted as the number of classes in Algorithm 1.

* The proposition (2) is problematic.
  * The $\epsilon$ should relate to $\gamma$ and $N$, otherwise, it sounds your approximator can achieve any $\epsilon$ under the current samples and probability tolerance. Additionally, why there is no $\gamma$ in the proof of proposition (2)?
  * As the authors pointed out $H$ is the Gaussian process which is approximated/replaced by the network $h$. Thus, there should be another term to quantify this approximation error, which is dependent on the width of the neural network. However, I cannot see anything implying this information in the theorems.

**Questions:**

* “If, on the other hand, some out-of-distribution input x is considered, then the error … is unlikely to be small” in Section 3.1: Even though seen $\boldsymbol x$ can push $g_{\theta^c}$ and $h_\phi$ to be closer by learning $\theta^c$, does this necessarily means that the difference $\| g\_{\theta^c}(x)-h\_\phi(x) \|$ for OOD $\boldsymbol x$ is unlikely to be small?
* Why does PEC have identical performances on different task splits as in Tables 1 and 2?

**Details Of Ethics Concerns:**

No.

---

> ### Author Response · Authors · 2023-11-16
> **Response, part 1/2**
>
> We thank the Reviewer for the review and helpful suggestions to improve our paper. We are glad that, in spite of the low overall score, the Reviewer considers our proposed method novel and that our extensive experiments support its good performance. Moreover, we feel that the weaknesses raised are not very severe and are appropriately addressed in our response.
>
> **The method ignores intrinsic interdependence among classes.**: We agree that this can be considered a limitation of our proposed method, and we clearly discussed this in Section 6 on page 9. However, the strong empirical performance of our method indicates that, in certain situations, ignoring interdependence among classes might be a great approach in practice.
>
> **The “efficiency” claimed in the contribution is less empirically supported.**: Firstly, typically PEC can achieve a given performance level with a lower number of epochs or parameters than other methods (e.g. most closely related VAE-GC), as shown in Figure 2 in our original submission. Secondly, to emphasize efficiency, we perform our main experiments in a challenging single-pass-through-data setting. Improvements of PEC over baselines are especially pronounced in this efficiency-focused setting and are typically larger than in multi-epoch settings, see Table 15 in the revised Appendix (Table 14 in the original submission). Finally, in the revised manuscript, Appendix B.2, paragraph “Number of operations needed”, we report the number of operations needed for inference in the case of PEC vs discriminative methods and show that here PEC is more efficient computation-wise in 4 of 5 considered cases.
>
> **Inconsistency in notation.**: We thank the Reviewer for pointing this out. In the revised manuscript we have made the notation consistent: we now consistently use $C$ to denote the number of classes and $N$ to denote the number of samples. We also made a few other changes (all indicated in orange in the revised manuscript) to improve the consistency in notation throughout the paper.
>
> **Proposition (2) is problematic.**
>
> ***The $\epsilon$ should relate to $\gamma$ and $N$.***: Rather than $\epsilon$ being a function of $\gamma$ and $N$, we just consider an appropriately powerful family of approximators (that depends on $\epsilon$, $\gamma$, and $N$) so that the assumptions are met. That means that to get better and better approximations, we would need to consider more powerful approximators (e.g., bigger neural networks).
>
> ***Why is there no $\gamma$ in the proof of Proposition (2)?***: There is $\gamma$ in the proof. Right before the inequality formulas, we look at the situation where two inequalities jointly hold, individual ones having probabilities at least $1 - \gamma$ and $1 - \delta$. We conclude that the joint event has the probability of at least $1 - \delta - \gamma$. We edited the proof so that this is more clearly explained.
>
> ***Should there be a term to quantify the error of approximating a GP $H$ by a network $h$?***: There is no term in Proposition 1 or Proposition 2 to quantify the approximation error of replacing the Gaussian Process $H$ by the network $h$ because in these two propositions, we connect GP-PEC($B$) to the GP posterior variance classification rule. In the paragraph above Proposition 1 (page 5), we in turn connect the practical method PEC to GP-PEC($B$) by setting $B=1$ and replacing the GP teacher $H$ with a neural network teacher $h$. We do not provide a theory that quantifies errors coming from that step (we clarify this in the revised Section 3.1). Instead, we justify this replacement by noting that wide neural networks with random initialization approximate GPs, and we provide empirical justification by showing that the performance of PEC improves by increasing the width of the teacher network but not by increasing the depth (see Section 4.5). Note that we do not claim that the theoretical section explains every aspect of our method. Rather, it provides additional motivation and interpretation of our approach. Supplementing this, we provide a strong empirical section demonstrating that our method works well in practice.
>
> **Is it guaranteed that the difference $|g_{\theta^c}(x)-h_{\phi}(x)|$ for OOD $x$ is unlikely to be small?**: Firstly, it has been demonstrated that neural networks struggle to generalize out-of-distribution [1], despite their good in-distribution performance; this gives intuitive support for our statement. Secondly, if we replaced here $h$ with a sample from a GP $H$, then Proposition 1 says that the expectation of this error is lower bounded by the variance of the GP posterior at the out-of-distribution data point. Such variance also should not be small because GP posterior only “learns” from the points that are distant to $x$.
>
> ***Note: this is part 1/2 of the response.***

---

> > ### Author Response · Authors · 2023-11-16
> > **Response, part 2/2**
> >
> > **Why does PEC have identical performances in Tables 1 and 2?**: The performance of PEC does not depend on the task split (i.e., how many classes there are in each task) because the class-specific student networks of PEC are only updated on examples from one class. Combined with the fact that we always use batch size = 1 for PEC, this means that it does not matter for PEC whether each task only contains one class (as in Table 1) or multiple classes (as in Table 2). This was explained in the last paragraph of Appendix B.2 (page 14), but we realize that we did not clearly refer to this explanation from the main text. In the revised manuscript we now refer to this explanation from the captions of both Table 2 and Figure 2.
> >
> > ***We thank the Reviewer again for their review. We believe we have adequately addressed all the issues raised by the Reviewer. Please let us know if there are any remaining issues, and otherwise please consider adapting the score of our paper.***
> >
> > *References*
> >
> > [1] Hendrycks, Dan, and Thomas Dietterich. "Benchmarking Neural Network Robustness to Common Corruptions and Perturbations." International Conference on Learning Representations. 2018.
> >
> > ***Note: this is part 2/2 of the response.***

---

> ### Author Response · Authors · 2023-11-20
> **Follow-up**
>
> We wanted to follow up on our initial response, and ask if the Reviewer is satisfied by how we addressed the raised issues.
>
> Please let us know if there are any further questions.

---

> > ### Comment · Reviewer_pyfp · 2023-11-21
> >
> > I appreciate the author’s further clarification and I am willing to raise the score by one level.
> >
> > That being said, I still have reservations about your oversimplified assumptions and theorems, where I cannot clearly see significant theoretical justification as you highlighted as one contribution. More concretely, on the one hand, the $\epsilon$ plays a similar role as the $\sigma_A^2$ in [1] (in contrast, your $\epsilon$ absorbs the value $M$ in the proof of Proposition 2 and this leads to a more stringent $\epsilon$) and it describes the intrinsic capacity of the hypothesis class. To obtain the smaller value of $\epsilon$, you need to require a wider network. On the other hand, the larger $M$ can sabotage the convergence rate of the second term in the proof of Proposition 2 (the very last line), where you just discard or incorporate it into big $\mathcal{O}$.
> >
> > Therefore, your rebuttal of “we just consider an appropriately powerful family of approximators…” is not that proper and indicates that you may overassume something.
> >
> > [1] Kamil Ciosek, Vincent Fortuin, Ryota Tomioka, Katja Hofmann, and Richard Turner. Conservative uncertainty estimation by fitting prior networks. In International Conference on Learning Representations, 2020.

---

> > > ### Author Response · Authors · 2023-11-23
> > >
> > > We thank the Reviewer for engaging with our rebuttal. We are glad that we have been able to address the main concerns of the Reviewer, and that they decided to raise their score.
> > >
> > > We also appreciate the further comments from the Reviewer regarding $\epsilon$ in Proposition 2. To make the assumption that underlies this proposition clearer, we have clarified that we assume that $h$ can be approximated by $g^{h,X}$ so that, ***for arbitrarily large training sets***, the training error is smaller than $\epsilon >0$ with probability at least $1-\gamma$ (all new changes for the 2nd rebuttal are indicated in blue in the revised manuscript). Consequently, the family of approximators only depends on the chosen $\epsilon$ and $\gamma$, not $N$. We note that this assumption is analogous to the one made in [1], Proposition 2, where it is assumed that the Bayes error can be reached for arbitrarily large training sets.
> > >
> > > Based on the Reviewer’s comments, we further changed the claim that our theory provides “justification” for PEC to that it provides “support” for PEC (e.g., by providing an additional motivation and interpretation). The main *justification* for our method comes from its strong empirical performance.
> > >
> > > [1] Kamil Ciosek, Vincent Fortuin, Ryota Tomioka, Katja Hofmann, and Richard Turner. Conservative uncertainty estimation by fitting prior networks. In International Conference on Learning Representations, 2020.

---

### Official Review · Reviewer_McZJ · 2023-10-30

**Soundness:** 3 good
**Presentation:** 3 good
**Contribution:** 3 good
**Rating:** 8
**Confidence:** 3

**Summary:**

The authors present PEC (Prediction Error-based Classification), a method for continual learning and class-incremental learning in which for each class, a student neural network learns to reproduce a randomly-initialized-and-then-frozen neural network only on examples from that class. Continual learning concerns supervised learning where data is not iid and may evolve. Task-based class incremental learning involves single-pass training in stages where only a few classes need to be discriminated between but detecting the union of all classes is required at test phase (in production).  The authors argue that the one-model-per-class approach of PEC prevents forgetting and interference between classes and can handle one-class-at-a-time-presented scenarios. They argue that using a neural net mapping is easier and simpler than creating a generative model for each class.   Theoretical connections between the method and gaussian processes are derived, given known results about the correspondence between very wide neural nets and gaussian processes. Mostly strong experimental results are presented in comparison with 11 competing approaches , although imbalanced training sets can give PEC some trouble

**Strengths:**

The paper is extremely well written and clear. The approach is original as far as I know, although I'm not an expert on continual learning. I've refereed for NeurIPS, ICML, ICLR and AAAI for the past several years, and this is one of the best written papers I've refereed.  The concept and motivation underpinning the approach are easy to understand and well presented. There is a nice dovetail between theory and experiment with the correspondence with gaussian processes, and it's particularly theoretically satisfying to see neural net width helping but depth not helping given the theoretical correspondence of GPs with wide neural nets. It's also a  nice counterweight to the prevailing research neural net trends of the past decade, in which the benefits of depth (the D in DNN), i.e. > 1 hidden layer, have been the main focus.

The experimental results look strong to me but also credible in their honesty, with PEC usually but not always winning.

**Weaknesses:**

My biggest concern with the paper is a lack of emphasis on or elucidation of a crucial aspect of the approach, namely, the details of the random parameter generation for the teacher neural network.  Algorithm 1 says 'initialize phi with random values' but doesn't say how. I had to dig into appendex B to figure out that default PyTorch initialization was used, i.e.,  Kaiming.   Those of us who have coded up neural nets completely from scratching using only low level programming in e.g. C++ or even Python with only numpy know well that the details of the initialization can make a big difference to the speed and/or success of training, certainly for sigmoidal activations and to some extent for ReLu/GeLu. If the neural net weights are too large relative the input magnitudes, you can get stuck in the saturated, near-constant regions of the activation function and the training error will barely move. If the weights are too small, you can get stuck for a long time in the linear region of the activation function and it can take a long time for training to implement any nonlinearity.  Now, it may well be that Kaiming initialization with GeLu or ReLU and then freezing will ~always result in a reasonable, useful teacher network which successfuly discriminates between classes, but this needs to be emphasized by the authors. I also wish they had done experiments to see how sensitive the results are to the initialization.

I also think this sentence on page 3 is wrong/confusing:

"As the architectures for g and h, we use shallow neural networks with one hidden layer, either
convolutional or linear, followed by a final linear layer"

When I read that in some cases the hidden layer is linear and the final layer is linear, I found it strange...that would mean the whole neural net is a linear mapping and it would seem surprising to me that a linear mapping would characterize the class well enough. However, in Table 5 on page 13, I saw that in fact there is a GeLu layer, so there is some nonlinearity. This should be clarified on page 3.

That brings up another suggestion: although architecture choices are explored in section 4.5, it would be a nice ablation-ish experiment to see how 0-hidden layers (i.e. purely linear logistic regression) does as a target teacher network.  Is the nonlinearity of the neural net actually important? The GP theory correspondence would suggest yes, but it would be nice to either confirm that or learn from a result of a linear teacher actually being successful,

Another minor point: I don't think the title conveys the core of the approach very well.  Technically, "prediction error" is indeed involved, but prediction error usually refers to a prediction you actually care about intrinsically in the real world rather than predicting a random neural net mapping.  Something along the lines of 'random neural net class modeling for continual learning' might be a better title. That's just my two cents, though, and I'm not even sure you're allowed to change the title.

I'm definitely open to boosting my score with a few tweaks to address concerns.

*** Update after reading rebuttal ***

I appreciate the additional experiments and clarifications regarding initialization and linear ablation studies. I note that random, small magnitude initialization does hurt performance (particularly MNIST) substantially, so I'm glad that initialization is now emphasized more. Accordingly, I have raised my score to an 8. However, I will point out to the Area Chair and any other meta-reviewers that my confidence is only a 3. Reviewer HWad seems to have more familiarity with continual learning and the concerns raised in that review make my slightly nervous. I wouldn't be prepared to fight vigorously for acceptance. Nonetheless, I did increase my score to 8.

**Questions:**

Do you have experiments on 0-hidden-layer teachers?

Do you have experiments on sensitivity to initialization? Can you emphasize the importance of sensible initialization in the main body of the paper? Not everyone will know to use an out-of-the-box reliably smart initialization like Kaiming which is designed to get the neural to be living in the "right" regions of the activation function.

---

> ### Author Response · Authors · 2023-11-16
>
> We thank the Reviewer for the encouraging review and helpful suggestions to improve our paper. We are glad that the Reviewer finds our paper well-written, our approach original, and the experimental results strong. Below we address individual suggestions and questions.
>
> **Insufficient discussion on initializations**: We agree that the initialization is a design choice that might be especially important for our method. We now report additional experiments in Appendix C.7 where we replace the Kaiming initialization with other choices, such as Xavier initialization and random uniform initializations with various supports. For convenience, we include the result table here as well:
>
> | Method                                   | MNIST         | CIFAR-10      |
> |------------------------------------------|---------------|---------------|
> | PEC, Kaiming (vanilla)                   | 92.31 (±0.13) | 58.94 (±0.09) |
> | PEC, Xavier                              | 91.70 (±0.13) | 58.02 (±0.14) |
> | PEC, random uniform from [−0.001, 0.001] | 88.52 (±0.22) | 57.33 (±0.22) |
> | PEC, random uniform from [−0.01, 0.01]   | 91.73 (±0.09) | 57.60 (±0.20) |
> | PEC, random uniform from [−0.1, 0.1]     | 92.01 (±0.14) | 57.46 (±0.20) |
>
> We note that PEC’s performance remains relatively stable across the studied options. In addition, we now also include information on the used initialization scheme in the main text, Section 3.1, pages 3/4 (we mark all revised fragments with orange).
> Finally, we note that in practice, implementations in popular libraries such as PyTorch or TensorFlow typically use Kaiming or Xavier initialization by default, so we expect most ML practitioners implementing our method to get good performance by default.
>
> **Wrong/confusing sentence about PEC architecture, page 3**: Thank you for spotting this. We revised the text to include the information that there is a GELU (when the hidden layer is linear) or a ReLU (when the hidden layer is convolutional) nonlinearity before the final linear layer.
>
> **PEC with linear architecture**: We include an additional experiment comparing vanilla PEC to PEC with linear networks. We consider two variants. For both variants, the described changes are applied to both the teacher and the student networks. In *Removed nonlinearity*, we take the vanilla PEC architecture and remove the nonlinear activation function, resulting in a linear network that retains some inductive biases of the original architecture (e.g. the use of convolutions). In *Single linear*, we use a single linear layer with the output size chosen to match the parameter count of the original architecture. We report the results in Appendix C.8. For convenience, we include the result table here as well:
>
> | Method                    | MNIST          | Balanced SVHN  | CIFAR-10       | CIFAR-100      | miniImageNet   |
> |---------------------------|----------------|----------------|----------------|----------------|----------------|
> | PEC                       | 92.31 (± 0.13) | 68.70 (± 0.16) | 58.94 (± 0.09) | 26.54 (± 0.11) | 14.90 (± 0.08) |
> | PEC, Removed nonlinearity | 92.41 (± 0.12) | 61.85 (± 0.20) | 46.19 (± 0.14) | 19.41 (± 0.11) | 10.55 (± 0.11) |
> | PEC, Single linear        | 80.39 (± 0.24) | 71.08 (± 0.18) | 40.04 (± 0.14) | 13.21 (± 0.10) | 7.30 (± 0.11)  |
>
> We see that indeed the variant with nonlinearity generally performs better, or even substantially better, than the linear variants. This is in correspondence to the GP theory. In some cases, the performance of the linear variant of PEC is still pretty strong (e.g., for Balanced SVHN – the reason might be that the convolutional architecture that we use for vanilla PEC might not be a good choice for this dataset).
>
> **Paper’s title**: We appreciate the suggestion and understand the argument; however, we would prefer to keep the name of the method and the title of the paper. The current name is a compromise between precision and conciseness, and we clarify the nature of the prediction error in the abstract.
>
> **We hope that we exhaustively answered the Reviewer’s questions. Please let us know if there are any remaining issues, and otherwise, we ask the Reviewer to consider raising the score.**

---

> > ### Comment · Reviewer_McZJ · 2023-11-20
> > **Thanks for the additional experiments**
> >
> > I appreciate the additional experiments and clarifications regarding initialization and linear ablation studies. I note that random, small magnitude initialization does hurt performance (particularly MNIST) substantially, so I'm glad that initialization is now emphasized more. Accordingly, I have raised my score to an 8. However, I will point out to the Area Chair and any other meta-reviewers that my confidence is only a 3. Reviewer HWad seems to have more familiarity with continual learning and the concerns raised in that review make my slightly nervous. I wouldn't be prepared to fight vigorously for acceptance. Nonetheless, I did increase my score to 8.

---

> > > ### Author Response · Authors · 2023-11-20
> > > **Thank you**
> > >
> > > We thank the Reviewer for engaging with our rebuttal. We are glad that the Reviewer appreciates our additional experiments and that they raised their score.

---

### Official Review · Reviewer_TNjc · 2023-10-31

**Soundness:** 4 excellent
**Presentation:** 4 excellent
**Contribution:** 3 good
**Rating:** 8
**Confidence:** 3

**Summary:**

The authors present a challenging variant of continual learning, known as Class-incremental learning (CIL). The goal in CIL is to learn to distinguish all classes that are introduced incrementally. Existing strategies often exhibit excessive forgetting and score imbalance for classes not seen together during training.

The authors introduce a novel approach, Prediction Error-based Classification (PEC). Unlike traditional paradigms, PEC calculates a class score by gauging the prediction error of a model trained to mirror the outputs of a static random neural network for that class. This method is likened to a classification rule based on Gaussian Process posterior variance.

PEC offers several practical benefits, such as sample efficiency, ease of tuning, and effectiveness even when data are presented class by class. Empirical results demonstrate that PEC performs robustly in single-pass-through-data CIL, outperforming other rehearsal-free baselines and, in most cases, also rehearsal-based methods with a moderate replay buffer size across multiple benchmarks.

Small issues:
In 4.4: "We hypothesize this is due to [...]" - Please re-read the sentence.

**Strengths:**

- Very well written and easy to follow, despite the complexity. The authors aim to provide intuitive explanations, without sacrificing formal clarity. Thanks!
- The method offers clear practical advantages of existing alternatives and good performance.
- Theoretical connections are drawn to substantiate the motivation.
- Empirical results are convincing.
- Experimental setup is described in depth.

**Weaknesses:**

- I previously saw this on Arxiv. This is not a double-blind review. I don't see any guidelines of how to deal with this. Hence, I at least want to mention it here.

**Questions:**

- In the intro it says: "Nonetheless, these methods typically perform worse with class-incremental learning than in the easier task-incremental setting" - How is task incremental easier than class incremental? Is it not simply a more general problem?
- You say: "[...] the teacher network’s middle layer is typically wider than the student’s one." Why?

---

> ### Author Response · Authors · 2023-11-16
>
> We thank the Reviewer for the encouraging review. We are glad that the Reviewer finds our paper well-written, considers our empirical results convincing, and appreciates the practical advantages of our method as well as the theoretical connections we draw. Below we address the individual suggestions and questions.
>
>
> **Confusing sentence in 4.4**: Thank you for pointing this out. In the updated manuscript we removed this confusing sentence and we made a few changes to other sentences in this subsection to improve clarity.
>
> **How is task incremental easier than class incremental? Is it not simply a more general problem?**: We agree that it is too simplistic to say that task-incremental learning is easier than class-incremental learning. Thank you for pointing this out. We corrected this in the updated manuscript.
>
> **Why the teacher network’s middle layer is typically wider than the student’s one?**: Firstly, we motivate our method as approximating the criterion based on GP variance. For this, the teacher network should approximate a GP, and it has been shown that neural networks approach GPs in the limit of infinite width [1]. Secondly, by having a wider teacher than the student, we are giving a “difficult task” to the student, hence it cannot just perfectly copy the weights from the teacher (which would be undesirable since the loss would be 0 everywhere). Finally, in Section 4.5 we empirically showed that having a wide teacher is beneficial for performance.
>
> *References*
>
> [1] Radford M Neal. Priors for infinite networks. Bayesian learning for neural networks, pp. 29–53, 1996.

---

> > ### Comment · Reviewer_TNjc · 2023-11-20
> > **Thanks for the response and clarification.**
> >
> > Thank you.

---

### Meta-Review · Area_Chair_19Nu · 2023-12-06

**Metareview:**

## Overview

This paper describes an approach to online class-incremental learning. The authors propose a method based on Prediction Error-based Classification (PEC) that uses class-specific student networks trained to replicate outputs of randomly initialized static networks. Classification performed by selecting the class with the smallest error between all the fitted models up to the current task and the output representation of the random prior network. The PEC approach is sample efficient and outperforms on average other exemplar-free approaches and a broad variety of benchmark datasets and scenarios.

## Strengths

Reviewers were generally quite positive about the proposed approach. Noting that the manuscript is very well-written and easy to follow despite the complexity of the proposed approach, and PEC has a number of practical advantages in addition to its good overall performance. All reviewers recognize the novelty of the proposed approach, with some drawing connections to the broader literature that could help the authors frame the contribution in the broader context of class-incremental learning. Finally, reviewers acknowledge the extensive and honest experimental comparison with the literature on online class-incremental learning.

## Weaknesses

Reviewers raised a number of points regarding the weaknesses of the paper. These generally ranged from simple clarifications about notation and technical details, to more serious questions about the thoroughness of experimental comparison with the literature on exemplar-free incremental learning, missing ablations, and a thorough analysis of space and time complexity. The authors engaged with these more serious concerns, providing clarifications and new experimental results (an ablation on ensembling versus PEC, an analysis of computational complexity, linear/nonlinear PEC comparison, an ablation on random initialization). The AC notes that these new results are *complementary* to the original manuscript rather than being fundamentally new results.

The main outstanding weaknesses after the rebuttal phase regard some remaining doubts regarding oversimplified assumptions underlying the theoretical foundations of the PEC approach, and some remaining questions regarding comprehensiveness of ablations and state-of-the-art comparisons. The authors defended themselves well, meeting where possible the reviewer requests somewhere in the middle. The AC finds significant value in this work and feels that it adds something interesting to the discussion on class-incremental learning.

**Justification For Why Not Higher Score:**

Several reviewers raised specific points regarding the thoroughness of ablations (partially mitigated in rebuttal) and theoretical assumptions. As such, it seems like the work makes for a strong poster presentation in its current form. That being said, the quality of the presentation and writing (commented almost universally by the reviewers) means that I would not be offended if it were bumped to Spotlight.

**Justification For Why Not Lower Score:**

Through the reviewer/author interactions the authors responded convincingly the the primary concerns of the reviewers. The remaining outstanding concerns regard comprehensiveness of experiments (to an unreasonable standard for a conference paper) and theoretical assumptions (effectively balanced by the empirical validation).

---

### Decision · Program_Chairs · 2024-01-16

Accept (poster)